# The Sharpness Disparity Principle in Transformers for Accelerating Language Model Pre-Training

**Jinbo Wang** [* 1] **Mingze Wang** [* 1] **Zhanpeng Zhou** [* 2] **Junchi Yan** [2] **Weinan E** [1 3 4] **Lei Wu** [1 3 4]

## Abstract

Transformers consist of diverse building blocks, such as embedding layers, normalization layers, self-attention mechanisms, and point-wise feed-forward networks. Thus, understanding the differences and interactions among these blocks is important. In this paper, we uncover a clear **sharpness disparity** across these blocks, which emerges early in training and intriguingly persists throughout the training process. Motivated by this finding, we propose **Blockwise Learning Rate (LR)**, a strategy that tailors the LR to each block's sharpness, accelerating large language model (LLM) pre-training. By integrating Blockwise LR into AdamW, we consistently achieve lower terminal loss and nearly $2\times$ speedup compared to vanilla AdamW. We demonstrate this acceleration across GPT-2 and LLaMA, with model sizes ranging from 0.12B to 2B and datasets of OpenWebText, MiniPile, and C4. Finally, we incorporate Blockwise LR into other optimizers such as Adam-mini (Zhang et al., 2024c), a recently proposed memory-efficient variant of Adam, achieving a combined $2\times$ speedup and $2\times$ memory saving. These results underscore the potential of exploiting the sharpness disparity to improve LLM training.

## 1. Introduction

Transformers (Vaswani et al., 2017) have achieved remarkable success across fields, including natural language processing (Brown et al., 2020), vision (Dosovitskiy et al., 2020), and scientific computing (Jumper et al., 2021).

---

[*]Equal contribution [1]School of Mathematical Sciences, Peking University [2]Shanghai Jiao Tong University [3]Center for Machine Learning Research, Peking University [4]AI for Science Institute, Beijing, China. Correspondence to: Lei Wu <leiwu@math.pku.edu.cn>, Mingze Wang <mingzewang@stu.pku.edu.cn>.

*Proceedings of the $42^{nd}$ International Conference on Machine Learning*, Vancouver, Canada. PMLR 267, 2025. Copyright 2025 by the author(s).

They have become the de facto design in modern AI models (Team et al., 2023; Achiam et al., 2023; Liu et al., 2024a).

Compared to traditional architectures, e.g., multilayer perceptrons (MLPs), convolutional neural networks (CNNs), and recurrent neural networks (RNNs), transformers exhibit distinctive **alloy-like characteristics**, where **diverse types of blocks** synergistically combine to achieve superior performance. A transformer at minimum includes self-attention (further broken down into query-key (QK) and value-output (VO)) blocks, point-wise feedforward networks (FFN), normalization layers (Norm), and embedding layers (Emb). Uncovering the distinct properties of these blocks, as well as the differences and interactions among them, is thus crucial for gaining a deeper insight into transformer models (Wang & E, 2024).

In practice, transformers are typically trained using the AdamW optimizer (Kingma & Ba, 2014; Loshchilov & Hutter, 2017). Dissecting the alloy-like characteristics of transformers can provide insights into why Adam outperforms stochastic gradient descent (SGD) for transformer training (Devlin, 2018; Zhang et al., 2020; Pesme & Flammarion, 2023; Kunstner et al., 2024; Zhang et al., 2024b) and even holds promise for unlocking further improvements in training efficiency (Popel & Bojar, 2018; Xiong et al., 2020; Zhang et al., 2024c). Particularly, Zhang et al. (2024b) and Zhang et al. (2024c) observed that unlike MLPs and CNNs, the Hessian (aka sharpness or curvature) of transformers exhibits a distinct blockwise heterogeneity. Building on this insight, Zhang et al. (2024c) successfully reduced Adam's memory footprint nearly by half without sacrificing training efficiency for a variety of LLM and non-LLM training tasks.

**Our Contribution.** In this work, we aim to explore how we can leverage the aforementioned alloy-like characteristics of transformers to improve training efficiency. Specifically, our contributions can be summarized as follows:

- **The sharpness disparity principle**. Motivated by the alloy-like characteristics, we examine the sharpness of transformers at the level of block type. Surprisingly, we discover a distinct disparity in sharpness across different block types, summarized as follows:

$$\mathcal{S}(\mathsf{Emb}) \ll \mathcal{S}(\mathsf{QK}) < \mathcal{S}(\mathsf{FFN}) < \mathcal{S}(\mathsf{VO}) \ll \mathcal{S}(\mathsf{Norm}) \quad (1)$$

Here $\mathcal{S}(\bullet)$ denotes the average sharpness of block type $\bullet$ (see Eq.(4) for the calculation details). See Figure 1 (left) for an illustration of this principle. Intriguingly, this principle emerges in the early training stage and persists throughout the subsequent training process, as shown in Figure 3. These findings are validated through extensive experiments on the training of GPT-2 (Radford et al., 2019) and LLaMA models (Touvron et al., 2023), spanning various model sizes and datasets. We also provide preliminary theoretical explanations to complement these empirical observations.

- **The Blockwise LR strategy.** Inspired by **Principle** (1), we propose tuning LRs by block type to accelerate LLM pre-training. Specifically, we adjust the LRs of blocks within the same type in proportion to their sharpness, while keeping the LR of the block type with the highest sharpness unchanged. This strategy accelerates the dynamics along low-sharpness directions without compromising training stability, as the latter is governed by the high-sharpness directions.

The effectiveness of Blockwise LR is extensively validated in LLM pre-training across both GPT-2 and LLaMA models, with model sizes ranging from 0.12B to 2B parameters, and datasets including OpenWebText (Gokaslan & Cohen, 2019), MiniPile (Kaddour, 2023), and C4 (Raffel et al., 2020). The results can be summarized as follows:

> AdamW with Blockwise LR achieves lower terminal loss and is nearly $2\times$ **faster** than vanilla AdamW.

See Figure 1 (right) for a quick view of the acceleration effect achieved by Blockwise LR. Furthermore, we explore the compatibility of Blockwise LR with other Adam-based optimizers. Specifically, we integrate our Blockwise LR into Adam-mini (Zhang et al., 2024c), achieving both $2\times$ speedup and $2\times$ memory saving.

**Remark 1.1.** *There has been a long-standing effort in deep learning to accelerate neural network training by adapting layerwise learning rates, a strategy that has proven effective in architectures such as MLPs and CNNs (Yang, 2019; Yang et al., 2022; Everett et al., 2024; Shin et al., 2024). However, these approaches have not been successfully transferred to the training of deep transformers. We hypothesize that this gap stems from transformers' distinctive alloy-like characteristics: the inherent block-level diversity makes layerwise learning rate strategies inadequate. To investigate this further, we examine layer-level sharpness in Figure 9 and no clear trends emerge across layers. This suggests that while sharpness disparity exists at the block-type level, it does not exhibit a consistent pattern at the layer level.*

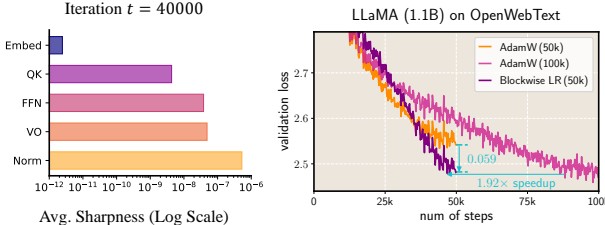

Figure 1: (left) Sharpness disparity among block types in a pre-trained GPT-2 (small) on OpenWebText, exhibiting a clear order relationship as characterized by **Principle** (1). (right) For the pre-training of LLaMA (1.1B) on OpenWebText, incorporating our Blockwise LR strategy into AdamW results in a lower terminal loss and a $1.92\times$ speedup compared to the well-tuned vanilla AdamW.

## 2. Related Works

**Sharpness structures in transformers.** Recent work has started to investigate blockwise sharpness patterns in transformer models through Hessian-based analyses. For example, Zhang et al. (2024b) empirically observed the sharpness' blockwise heterogeneity but did not establish a clear principle regarding the sharpness disparity among different blocks. Meanwhile, Ormaniec et al. (2024) provided a Hessian analysis for a single self-attention (SA) layer, focusing only on the sharpness disparity between the query-key (QK) and value-output (VO) blocks within the same layer.

In contrast, we examine sharpness at the block-type level across the entire transformer architecture, rather than focusing on individual blocks (as in Zhang et al. (2024b)) or a single layer (as in Ormaniec et al. (2024)). This coarse-grained perspective reveals a consistent disparity, as formalized by **Principle** (1), which persists throughout most of the training process—except during the initial steps.

**Efficient optimizers for LLM pre-training.** AdamW (Adam with decoupled weight decay) (Loshchilov & Hutter, 2017) has become the default optimizer in LLM pre-training. Efforts to design more efficient optimizers generally fall into two main categories: accelerating convergence and reducing memory footprint. Accelerations have been developed using techniques such as Nesterov momentum (Xie et al., 2022), diagonal second-order estimates (Liu et al., 2024b; Wang et al., 2024), variance reduction (Yuan et al., 2024), and matrix-based preconditioners (Keller et al., 2024; Vyas et al., 2024). Memory-efficient optimizers utilize sign-based methods (Chen et al., 2024), reduced usage of second moments in Adam (Zhang et al., 2024c), and gradient low-rank projection (Zhao et al., 2024a). The closest work to our Blockwise LR is Wang et al. (2024), which also increases the LR along low-sharpness directions. A detailed comparison is deferred to Section 5.

**The edge of stability (EoS) phenomenon.** Neural network

training typically occurs at the EoS stage (Wu et al., 2018; Jastrzebski et al., 2020; Cohen et al., 2021; 2022), where the optimizer exhibits oscillatory behavior along sharp directions without diverging, while steadily progressing along flat directions, leading to loss reduction. Several works (Wen et al., 2024; Song et al., 2024; Cohen et al., 2024; Wang et al., 2024) have highlighted the crucial role of the dynamics along flat directions (referred to as river directions by Wen et al. (2024), bulk directions by Song et al. (2024), and stable direction in Wang et al. (2024)) in reducing total loss. Notably, Wen et al. (2024) further demonstrated that this picture is essential for understanding LLM pre-training. Building on these insights, our Blockwise LR approach is designed to accelerate training by amplifying the dynamics particularly along the flat river directions.

## 3. Preliminaries

**Notations.** Let $\|\cdot\|_2$, $\|\cdot\|_F$, and $\mathrm{Tr}(\cdot)$ denote the spectral norm, Frobenius norm and trace for matrices, respectively. Given $\boldsymbol{A} \in \mathbb{R}^{m \times n}$, its row-wise vectorization is defined as $\mathrm{vec}(\boldsymbol{A}) = (a_{1,1}, \cdots, a_{1,n}, \cdots, a_{m,1}, \cdots, a_{m,n}) \in \mathbb{R}^{mn}$. The Kronecker product and Hadamard product are denoted by $\otimes$ and $\odot$, respectively. The row-wise mean and covariance of $\boldsymbol{A} \in \mathbb{R}^{m \times n}$ are denoted by $\mathbb{E}_r[\boldsymbol{A}] \in \mathbb{R}^{m \times n}$ and $\mathbb{V}_r[\boldsymbol{A}] \in \mathbb{R}^{m \times n}$, respectively. Specifically, they are defined as: for all $i \in [m], j \in [n]$, $(\mathbb{E}_r[A])_{i,j} = \frac{1}{n}\sum_{k=1}^n A_{i,k}$, $(\mathbb{V}_r[A])_{i,j} = \left(A_{i,j} - \frac{1}{n}\sum_{k=1}^n A_{i,k}\right)^2$. We will use standard big-O notations like $\mathcal{O}(\cdot)$, $\Omega(\cdot)$, and $\Theta(\cdot)$ to hide problem-independent constants.

**Jacobian matrix.** Given a vector-valued function: $\boldsymbol{b} \mapsto \boldsymbol{a}(\boldsymbol{b})$ with $\boldsymbol{b} \in \mathbb{R}^n$ and $\boldsymbol{a}(\boldsymbol{b}) \in \mathbb{R}^m$, the Jacobian is defined as $\frac{\partial \boldsymbol{a}}{\partial \boldsymbol{b}} = (\frac{\partial a_i}{\partial b_j})_{i,j} \in \mathbb{R}^{m \times n}$. Analogously, for a matrix-valued function: $\boldsymbol{B} \mapsto \boldsymbol{A}(\boldsymbol{B})$ where $\boldsymbol{B} \in \mathbb{R}^{p \times q}$ and $\boldsymbol{A}(\boldsymbol{B}) \in \mathbb{R}^{m \times n}$, to avoid directly working with tensors, the Jacobian is defined as $\frac{\partial \boldsymbol{A}}{\partial \boldsymbol{B}} := \frac{\partial \mathrm{vec}(\boldsymbol{A})}{\partial \mathrm{vec}(\boldsymbol{B})} \in \mathbb{R}^{mn \times pq}$.

### 3.1. The Transformer Architecture

Given an $n$-token input sequence $\boldsymbol{X} = (\boldsymbol{x}_1^\top, \cdots, \boldsymbol{x}_n^\top)^\top \in \mathbb{R}^{n \times d}$, where $d$ refers to the vocabulary size in LLM and each $\boldsymbol{x}_i$ corresponds to the token's one-hot encoding, an $L$-layer transformer $\mathsf{TF}$ processes it as follows.

**Embedding layer.** First, each input token is embedded into the latent space through an embedding layer with parameters $\boldsymbol{W}_E \in \mathbb{R}^{d \times D}, \boldsymbol{b}_E \in \mathbb{R}^{1 \times D}$:

$$\boldsymbol{x}_s^{(0)} = \boldsymbol{x}_s \boldsymbol{W}_E + \boldsymbol{b}_E, \ s \in [n],$$

where the bias $\boldsymbol{b}_E$ is omitted in LLMs such as nanoGPT (Karpathy, 2022).

**$L$-layer SA-FFN blocks.** Then the embedded sequence $\boldsymbol{X}^{(0)}$ is processed by $L$-layer SA-FFN blocks, and the output of the final layer is taken as the output sequence

$\mathsf{TF}(X) = X^{(L)} \in \mathbb{R}^{n \times D}$. For each layer $l \in [L]$, the computations are as follows:

$$\begin{aligned} \boldsymbol{X}^{(l-\frac{1}{2})} &= \boldsymbol{X}^{(l-1)} + \mathsf{SA}^{(l)}(\mathsf{Norm}^{(l-1/2)}(\boldsymbol{X}^{(l-1)})); \\ \boldsymbol{X}^{(l)} &= \boldsymbol{X}^{(l-\frac{1}{2})} + \mathsf{FFN}^{(l)}(\mathsf{Norm}^{(l)}(\boldsymbol{X}^{(l-\frac{1}{2})})). \end{aligned} \quad (2)$$

**Norm blocks.** Here, $\mathsf{Norm}^{(v)}$ ($v \in \{l - 1/2, l\}$) denote normalization layers (e.g., LayerNorm (Lei Ba et al., 2016) and RMSNorm (Zhang & Sennrich, 2019)) with learnable parameters $\boldsymbol{\gamma}^{(v)}, \boldsymbol{\beta}^{(v)} \in \mathbb{R}^{1 \times D}$. For LayerNorm, the computation for a token $\boldsymbol{x} \in \mathbb{R}^{1 \times D}$ is:

$$\mathsf{Norm}^{(v)}(\boldsymbol{x}) = \frac{\boldsymbol{x} - \mathbb{E}_r[\boldsymbol{x}]}{\mathbb{V}_r[\boldsymbol{x}]} \odot \boldsymbol{\gamma}^{(v)} + \boldsymbol{\beta}^{(v)}.$$

where the bias $\boldsymbol{\beta}$ is omitted in LLMs such as nanoGPT.

**FFN blocks.** $\mathsf{FFN}^{(l)}$ denotes a (token-wise) two-layer FFN of width $M$, comprising parameters $\boldsymbol{W}_1^{(l)} \in \mathbb{R}^{D \times M}, \boldsymbol{W}_2^{(l)} \in \mathbb{R}^{M \times D}$, and using activation function $\sigma(\cdot)$ such as ReLU. For any token $\boldsymbol{x} \in \mathbb{R}^{1 \times D}$, the operation is:

$$\mathsf{FFN}^{(l)}(\boldsymbol{x}) = \sigma(\boldsymbol{x}\boldsymbol{W}_1^{(l)})\boldsymbol{W}_2^{(l)}.$$

**SA blocks.** $\mathsf{SA}^{(l)}$, a multi-head self-attention, has parameters $\boldsymbol{W}_Q^{(l)}, \boldsymbol{W}_K^{(l)}, \boldsymbol{W}_V^{(l)}, \boldsymbol{W}_O^{(l)} \in \mathbb{R}^{D \times D}$. When applied to a sequence $\boldsymbol{Z} \in \mathbb{R}^{n \times D}$, it operates as:

$$\mathsf{SA}^{(l)}(\boldsymbol{Z}) = \sum_{h=1}^H \mathsf{SA}^{(l,h)}(\boldsymbol{Z})\boldsymbol{W}_O^{(l,h)}, \quad \mathsf{SA}^{(l,h)}(\boldsymbol{Z}) =$$

$$\mathrm{softmax}\left(\frac{\left\langle \boldsymbol{Z}\boldsymbol{W}_Q^{(l,h)}, \boldsymbol{Z}\boldsymbol{W}_K^{(l,h)} \right\rangle + \boldsymbol{M}}{\sqrt{D/H}}\right)\left(\boldsymbol{Z}\boldsymbol{W}_V^{(l,h)}\right),$$

where $H$ is the head number, and $\boldsymbol{W}_Q^{(l,h)}, \boldsymbol{W}_K^{(l,h)}, \boldsymbol{W}_V^{(l,h)} \in \mathbb{R}^{D \times (D/H)}$, $\boldsymbol{W}_O^{(l,h)} \in \mathbb{R}^{(D/H) \times D}$ are split from $\boldsymbol{W}_Q^{(l)}, \boldsymbol{W}_K^{(l)}, \boldsymbol{W}_V^{(l)}, \boldsymbol{W}_O^{(l)}$ by heads, respectively. The operator $\mathrm{softmax}(\cdot)$ represents the row-wise softmax normalization. For the next-token prediction, the mask $\boldsymbol{M} \in \mathbb{R}^{n \times n}$ satisfies $M_{i,j} = -\infty$ if $i < j$ and $M_{i,j} = 0$ otherwise.

### 3.2. Blockwise Sharpness and the Efficient Estimation

Measuring sharpness requires accessing the Hessian matrix, which is computationally expensive due to the high dimensionality of the parameter space. Consequently, approximate methods are needed to reduce computational complexity.

Let $\ell(\cdot, \cdot)$ denote the cross-entropy loss. For an input data $\boldsymbol{x} \in \mathbb{R}^{d_x}$ and label $\boldsymbol{y} \in \mathbb{R}^{d_y}$, let the model's prediction be $f(\boldsymbol{x}; \boldsymbol{\theta}) \in \mathbb{R}^{d_y}$. The Fisher (Gauss-Newton) matrix $F(\boldsymbol{\theta})$ is widely recognized approximation of the Hessian, particularly near minima. Thus, the diagonal Hessian can be estimated as $\boldsymbol{h}(\boldsymbol{\theta}) = \mathrm{diag}(F(\boldsymbol{\theta}))$, a popular technique in deep learning optimization (Martens & Grosse, 2015; Grosse &

Martens, 2016; George et al., 2018; Mi et al., 2022; Liu et al., 2024b; Wang et al., 2024). Moreover, given an input batch $\{(\boldsymbol{x}_b, \boldsymbol{y}_b)\}_{b=1}^B$, the empirical diagonal Fisher can be estimated: $\mathrm{diag}(\hat{F}(\boldsymbol{\theta})) = \frac{1}{B}\sum_{b=1}^B \nabla\ell(f(\boldsymbol{x}_b; \boldsymbol{\theta}); \hat{\boldsymbol{y}}_b) \odot \nabla\ell(f(\boldsymbol{x}_b; \boldsymbol{\theta}); \hat{\boldsymbol{y}}_b)$, where $\hat{\boldsymbol{y}}_b \sim \mathrm{softmax}(f(\boldsymbol{\theta}; \boldsymbol{x}_b))$. However, as noted by Liu et al. (2024b), implementing this estimator is computationally expensive due to the need to calculate $B$ single-batch gradients. Liu et al. (2024b) proposed a more convenient estimator $\mathrm{diag}(\hat{F}_{\mathrm{eff}}(\boldsymbol{\theta}))$, which only requires the computation of the mini-batch gradient $\nabla\hat{\mathcal{L}}_B(\boldsymbol{\theta}) = \frac{1}{B}\sum_{b=1}^B \nabla\ell(f(\boldsymbol{x}_b; \boldsymbol{\theta}); \hat{\boldsymbol{y}}_b)$ with $\hat{\boldsymbol{y}}_b \sim \mathrm{softmax}(f(\boldsymbol{x}_b; \boldsymbol{\theta}))$:

$$\boldsymbol{h}(\boldsymbol{\theta}) = \mathrm{diag}(\hat{F}_{\mathrm{eff}}(\boldsymbol{\theta})) = B \cdot \nabla\hat{\mathcal{L}}_B(\boldsymbol{\theta}) \odot \nabla\hat{\mathcal{L}}_B(\boldsymbol{\theta}). \quad (3)$$

According to Liu et al. (2024b, Section 2), this estimator is unbiased, i.e., $\mathbb{E}_{\hat{\boldsymbol{y}}}[\mathrm{diag}(\hat{F}_{\mathrm{eff}}(\boldsymbol{\theta}))] = \mathbb{E}_{\hat{\boldsymbol{y}}}[\mathrm{diag}(\hat{F}(\boldsymbol{\theta}))]$.

Given a block type $\bullet \in \{\mathsf{Emb}, \mathsf{QK}, \mathsf{VO}, \mathsf{FFN}, \mathsf{Norm}\}$, let $\boldsymbol{\theta}[\bullet]$ represent the parameters associated with all blocks of that type, and let $\boldsymbol{h}(\boldsymbol{\theta}[\bullet])$ denote the corresponding diagonal Hessian. The average sharpness for each block type can then be approximated as follows:

$$\mathcal{S}(\boldsymbol{\theta}[\bullet]) := \frac{\mathrm{Tr}(\boldsymbol{h}(\boldsymbol{\theta}[\bullet]))}{\#(\boldsymbol{\theta}[\bullet])} = \frac{B\left\|\nabla_{\boldsymbol{\theta}[\bullet]}\hat{\mathcal{L}}_B(\boldsymbol{\theta})\right\|_{\mathrm{F}}^2}{\#(\boldsymbol{\theta}[\bullet])}, \quad (4)$$

where $\hat{\mathcal{L}}_B$ corresponds to (3) and $\#(\boldsymbol{\theta}[\bullet])$ denotes the number of parameters associated with the block type $\bullet$. For brevity, $\boldsymbol{\theta}$ in (4) will be omitted when there is no ambiguity.

**Remark 3.1.** *It is worth noting that in (4), the sharpness is averaged over all blocks of the same type, which may be distributed across different layers, rather than being calculated within each individual block.*

## 4. The Sharpness Disparity Principle

### 4.1. Main Findings

We first investigate the sharpness disparity across different types of building blocks ($\mathsf{Emb}$, $\mathsf{QK}$, $\mathsf{VO}$, $\mathsf{FFN}$, $\mathsf{Norm}$) in transformer-based LLMs. Specifically, we pre-trained GPT-2 (Radford et al., 2019) and LLaMA (Touvron et al., 2023) models on the OpenWebText dataset using default configurations. Blockwise diagonal Hessians are analyzed at various checkpoints using the Hessian estimator (3). The experimental details can be found in Appendix A.1.

In Figures 1 (left) and 2 (left), we report the average sharpness, estimated using (4), of the five typical types of blocks for GPT-2 and LLaMA, respectively. The results reveal a clear and consistent sharpness disparity among different block types, as summarized in **Principle** (1). Specifically, $\mathsf{Norm}$ layers consistently exhibit the highest sharpness, the $\mathsf{Emb}$ layers are the flattest, and $\mathsf{QK}$ layers are relatively flatter compared to $\mathsf{FFN}$ and $\mathsf{VO}$ layers. These findings, to

the best of our knowledge, provide the first comprehensive comparison of sharpness across block types in transformers.

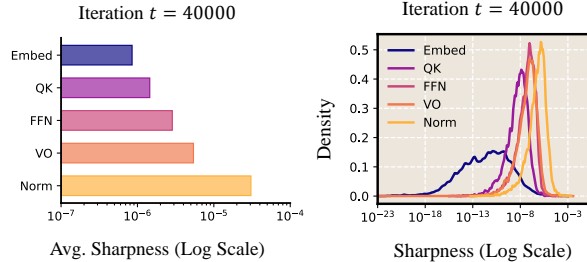

Figure 2: (left) The average sharpness for the five typical block types in a pre-trained LLaMA model (0.25B); (right) the sharpness distribution across different blocks in a pre-trained GPT-2 (small) model.

Figure 2 (right) plots the full sharpness distribution for each block type, whereas Figures 1 (left) and 2 (left) only report mean sharpness values. Evidently, even at the distribution level, **Principle** (1) remains valid. Interestingly, the $\mathsf{Emb}$ block exhibits much higher variance compared to other blocks. This behavior likely stems from the embedding layer's direct interaction with the entire vocabulary, where rare tokens result in the wide spread of small sharpness and frequent tokens contribute to large sharpness. A similar insight has been utilized by Kunstner et al. (2024) to explain the necessity of Adam in training NLP models.

Furthermore, Figure 3 illustrates the evolution of blockwise sharpness during the training process. We can see that **Principle** (1) is not exclusive to well-trained transformers; instead, it emerges in the early stages of training and persists consistently throughout the subsequent training process. This observation underscores the potential of leveraging **Principle** (1) to enhance LLM pre-training; we refer to Section 5 for further explorations.

**Comparison with existing works**. Our findings build on prior work, extending key observations. Zhang et al. (2024b) noted the block heterogeneity in the Hessian of transformers but did not establish a clear principle for sharpness distinctions across blocks, as we do with **Principle** (1). The work of Ormaniec et al. (2024) is more closely related but focuses solely on a single self-attention layer ($\mathsf{SA}$), reporting the relationship $\mathcal{S}(\mathsf{QK}) < \mathcal{S}(\mathsf{VO})$. In contrast, we analyze all major block types in transformers, including $\mathsf{Emb}$, $\mathsf{FFN}$, and $\mathsf{Norm}$, thereby offering a more comprehensive principle that captures the full scope of sharpness disparity.

### 4.2. Theoretical Insights

To provide theoretical insights into explaining **Principle** (1), we derive analytic expressions of $\mathcal{S}(\bullet)$ and analyze their dependence on parameter magnitudes and numbers of each block. For simplicity, we denote $\mathcal{Q}(\boldsymbol{\theta}) := \hat{\mathcal{L}}_B(\boldsymbol{\theta})$, where

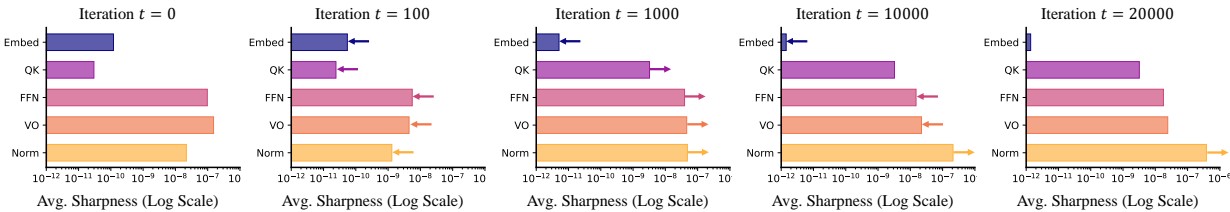

(a) Evolution of the average sharpness across different blocks during pre-training GPT-2 (small) on OpenWebText.

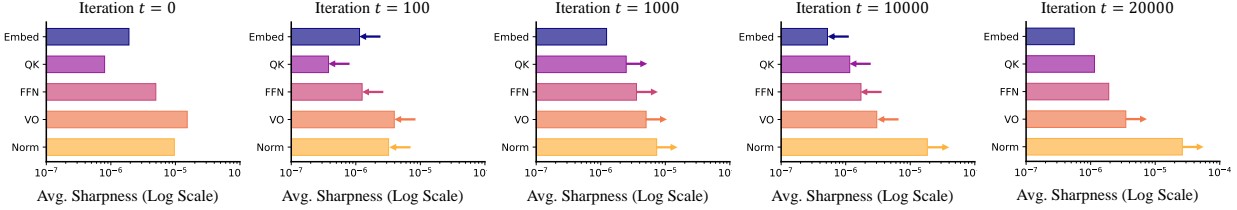

(b) Evolution of the average sharpness across different blocks during pre-training LLaMA (0.25B) on OpenWebText.

Figure 3: In these experiments, the total training steps are both 50k. **Principle** (1) emerges during the initial phase (from iteration 0 to iteration 1k), which accounts for only approximately $2\%$ of the total steps, and persists throughout the subsequent training process.

$\hat{\mathcal{L}}_B(\boldsymbol{\theta})$ is defined in (3). Then from (4), we have $\mathcal{S}(\bullet) = B \left\| \nabla_\bullet \mathcal{Q} \right\|_{\mathrm{F}}^2 / \#(\bullet)$. Without loss of generality, we set $B = 1$. Our calculations for $\nabla \mathcal{Q}$ apply to general $\mathcal{Q}$.

Considering blocks across different layers is complicated. Therefore, we focus on comparisons within the same layer. Specifically, we examine the following sharpness comparisons: (i) FFN vs. Norm within the same layer; (ii) SA (comprising QK and VO) vs. Norm within the same layer; and (iii) Emb vs. the adjacent Norm.

**Theorem 4.1** (FFN vs. Norm). *Consider the $l$-th layer in a transformer* (2). *Omitting the layer index for simplicity, let* $\boldsymbol{Y} = \boldsymbol{X} + \mathsf{FFN}\left(\mathsf{Norm}\left(\boldsymbol{X}; \boldsymbol{\gamma}\right); \boldsymbol{W}_1, \boldsymbol{W}_2\right)$, *where* FFN *utilizes the (Leaky) ReLU activation* $\sigma$. *Then, the gradients of* $\mathcal{Q}$ *w.r.t.* $\boldsymbol{W}_1, \boldsymbol{W}_2$, *and* $\boldsymbol{\gamma}$ *are:*

$$\frac{\partial \mathcal{Q}}{\partial \boldsymbol{W}_2} = \frac{\partial \mathcal{Q}}{\partial \boldsymbol{Y}}\left(\boldsymbol{X}_{\mathsf{Norm}}\boldsymbol{W}_1 \odot \frac{\partial \mathsf{A}}{\partial \mathsf{M}}\right) \otimes \boldsymbol{I}_d;$$

$$\frac{\partial \mathcal{Q}}{\partial \boldsymbol{W}_1} = \frac{\partial \mathcal{Q}}{\partial \boldsymbol{Y}}\left(\boldsymbol{I}_n \otimes \boldsymbol{W}_2^\top\right)\frac{\partial \mathsf{A}}{\partial \mathsf{M}}\left(\boldsymbol{X}_{\mathsf{Norm}} \otimes \boldsymbol{I}_M\right);$$

$$\frac{\partial \mathcal{Q}}{\partial \boldsymbol{\gamma}} = \frac{\partial \mathcal{Q}}{\partial \boldsymbol{Y}}\left(\boldsymbol{I}_n \otimes \boldsymbol{W}_2^\top\right)\frac{\partial \mathsf{A}}{\partial \mathsf{M}}\left(\boldsymbol{I}_n \otimes \boldsymbol{W}_1^\top\right)$$
$$\mathrm{diag}\left(\mathrm{vec}(\boldsymbol{X}_{\mathrm{std}})\right)\left(\mathbf{1}_{n \times 1} \otimes \boldsymbol{I}_d\right),$$

*where* $\boldsymbol{X}_{\mathrm{std}} := \frac{\boldsymbol{X} - \mathbb{E}_r[\boldsymbol{X}]}{\sqrt{\mathbb{V}_r[\boldsymbol{X}]}}, \boldsymbol{X}_{\mathsf{Norm}} := \mathsf{Norm}(\boldsymbol{X}; \boldsymbol{\gamma}) = \boldsymbol{X}_{\mathrm{std}} \odot \left(\mathbf{1}_{n \times 1} \otimes \boldsymbol{\gamma}\right), \mathsf{A} := \sigma(\mathsf{M}), \mathsf{M} := \boldsymbol{X}_{\mathsf{Norm}}\boldsymbol{W}_1$. *Let* $\Psi := n\sqrt{D}\left\|\frac{\partial \mathcal{Q}}{\partial \boldsymbol{Y}}\right\|_{\mathrm{F}}\left\|\frac{\partial \mathsf{A}}{\partial \mathsf{M}}\right\|_{\mathrm{F}}\left\|\boldsymbol{W}_1\right\|_{\mathrm{F}}\left\|\boldsymbol{W}_2\right\|_{\mathrm{F}}\left\|\boldsymbol{\gamma}\right\|_{\mathrm{F}}$. *Then, the blockwise average sharpness can be bounded as:*

$$\mathcal{S}(\boldsymbol{W}_\bullet) = \mathcal{O}\left(\frac{\Psi^2}{D^2 \|\boldsymbol{W}_\bullet\|_{\mathrm{F}}^2}\right), \bullet \in \{1, 2\};$$

$$\mathcal{S}(\boldsymbol{\gamma}) = \mathcal{O}\left(\frac{\Psi^2}{D \|\boldsymbol{\gamma}\|_{\mathrm{F}}^2}\right),$$

*where the denominators ($D^2$ or $D$) reflect the number of parameters in each group.*

Theorem 4.1 provides theoretical support for our main finding: $\mathcal{S}(\mathsf{FFN})$ is substantially smaller than $\mathcal{S}(\mathsf{Norm})$. As illustrated in Figure 8 (a), during training, $\|\boldsymbol{\gamma}\|_{\mathrm{F}}$ gradually decreases, and $\|\boldsymbol{W}_\bullet\|_{\mathrm{F}}$ ($\bullet \in \{1, 2\}$) in FFN layers remains larger than $\|\boldsymbol{\gamma}\|_{\mathrm{F}}$, resulting in $D^2 \|\boldsymbol{W}_\bullet\|_{\mathrm{F}}^2 \gg D \|\boldsymbol{\gamma}\|_{\mathrm{F}}^2$.

**Theorem 4.2** (QK, VO vs. Norm). *Consider the $(l - \frac{1}{2})$-th layer in* (2). *Omitting the layer index for simplicity, let* $\boldsymbol{Y} = \boldsymbol{X} + \mathsf{SA}\left(\mathsf{Norm}\left(\boldsymbol{X}; \boldsymbol{\gamma}\right); \boldsymbol{W}_K, \boldsymbol{W}_Q, \boldsymbol{W}_V, \boldsymbol{W}_O\right)$. *Consider a single-head attention (i.e., $H = 1$) for simplicity. Then, the gradients of $\mathcal{Q}$ w.r.t. different blocks $(\boldsymbol{W}_K, \boldsymbol{W}_Q, \boldsymbol{W}_V, \boldsymbol{W}_O, \boldsymbol{\gamma})$ are provided in Appendix B.2. Furthermore, there exist two problem-dependent constants $\Phi, \Psi > 0$ (detailed in Appendix B.2), such that:*

$$\mathcal{S}(\boldsymbol{W}_\bullet) = \mathcal{O}\left(\frac{\Phi^2}{D^2 \|\boldsymbol{W}_\bullet\|_{\mathrm{F}}^2}\right), \bullet \in \{K, Q\};$$

$$\mathcal{S}(\boldsymbol{W}_\bullet) = \mathcal{O}\left(\frac{\Psi^2}{D^2 \|\boldsymbol{W}_\bullet\|_{\mathrm{F}}^2}\right), \bullet \in \{V, O\};$$

$$\mathcal{S}(\boldsymbol{\gamma}) = \mathcal{O}\left(\frac{\Phi^2 + \Psi^2}{D \|\boldsymbol{\gamma}\|_{\mathrm{F}}^2}\right).$$

*where the denominators ($D^2$ or $D$) reflect the number of parameters in each group.*

Theorem 4.2 provides theoretical support for our main finding that both $\mathcal{S}(\mathsf{QK})$ and $\mathcal{S}(\mathsf{VO})$ are significantly smaller than $\mathcal{S}(\mathsf{Norm})$. The inclusion of the softmax operation in attention layers introduces additional complexity in the calculations. Detailed derivations are given in the appendix. As shown in Figure 8 (b), during training, $\|\boldsymbol{\gamma}\|_{\mathrm{F}}$ gradually

decreases, and $\|\boldsymbol{W}_\bullet\|_F$ ($\bullet \in \{K, Q, V, O\}$) in SA blocks remains larger than $\|\boldsymbol{\gamma}\|_F$, resulting in $D^2 \|\boldsymbol{W}_\bullet\|_F^2 \gg D \|\boldsymbol{\gamma}\|_F^2$.

This theorem does not explicitly establish that $\mathcal{S}(\mathsf{QK}) < \mathcal{S}(\mathsf{VO})$. Studying this relation requires a deeper analysis of the constants $\Phi$ and $\Psi$, as well as the magnitudes of $\|\boldsymbol{W}_\bullet\|_F$. Ormaniec et al. (2024) has demonstrated $\mathcal{S}(\mathsf{QK}) < \mathcal{S}(\mathsf{VO})$ both theoretically and experimentally, and we defer to that analysis instead of repeating it here.

**Theorem 4.3** (Emb v.s. Norm). *Consider the embedding layer and its adjoint normalization layer of a transformer* (2). *Omitting the layer index for simplicity, let:* $\boldsymbol{Y} := \mathsf{Norm}(\boldsymbol{X}\boldsymbol{W}_{\mathrm{emb}}; \boldsymbol{\gamma})$. *The gradients of $\mathcal{Q}$ w.r.t $\boldsymbol{W}_{\mathrm{emb}}$ and $\boldsymbol{\gamma}$ are derived in Appendix B.3. Moreover, there exists a problem-dependent constant $\Psi > 0$ (also detailed in Appendix B.3), such that:*

$$\mathcal{S}(\boldsymbol{W}_E) = \mathcal{O}\left(\frac{\Psi^2}{Dd \min_{i \in [d]} \|\tilde{\boldsymbol{w}}_{E_i}\|_2^2}\right);$$

$$\mathcal{S}(\boldsymbol{\gamma}) = \mathcal{O}\left(\frac{\Psi^2}{D \|\boldsymbol{\gamma}\|_F^2}\right),$$

*where $\tilde{\boldsymbol{W}}_E = (\tilde{\boldsymbol{w}}_{E_1}^\top, \cdots, \tilde{\boldsymbol{w}}_{E_d}^\top)^\top := \boldsymbol{W}_E - \mathbb{E}_r[\boldsymbol{W}_E]$. The denominators ($Dd$ or $D$) represent the number of parameters in each group.*

Theorem 4.3 provides theoretical justification for our main finding that $\mathcal{S}(\mathsf{Emb})$ is much smaller than $\mathcal{S}(\mathsf{Norm})$. As shown in Figure 8(c), during training, $Dd \|\tilde{\boldsymbol{w}}_{E_i}\|_2^2 \gg D \|\boldsymbol{\gamma}\|_F^2$. (Notice that the vocabulary size $d$ is very large in practice, e.g., 50304 for the GPT tokenizer.)

Recalling the definition of average sharpness (4), the key step in deriving Theorem 4.1 and 4.2, and 4.3 is establishing $\|\nabla_\bullet \mathcal{Q}\| = \mathcal{O}(1/\|\boldsymbol{\theta}[\bullet]\|)$. This relationship is highly intuitive given the compound multiplicative nature of transformer blocks, where the norm of the derivatives is inversely proportional to the norm of associated parameters, even with weak non-linearities. For example, if $y = \prod_{i=1}^n x_i$ and $\mathcal{Q} = \varphi(y)$, then $|\partial \mathcal{Q}/\partial x_i| = |\phi'(y)y/x_i| \propto 1/|x_i|$ for all $i \in [n]$.

## 5. The Blockwise LR Strategy

Recalling Figure 3, the sharpness disparity across different blocks, as described in (1), emerges early in training and persists until convergence. This insight can be leveraged to accelerate LLM pre-training, as elaborated later.

**Fast-slow dynamics at EoS.** As discussed in Section 2, recent studies (Wen et al., 2024; Song et al., 2024; Wang et al., 2024) have highlighted the distinct roles of the dynamics along high- and low-sharpness directions during EoS. The main picture is summarized as follows:

- **Fast dynamics**: Along *high-sharpness directions*, the optimizer exhibits significant fluctuations without converging or diverging. These components of dynamics govern training stability, as further increasing the LR in these directions can lead to instability, while contributing little to loss reduction.

- **Slow dynamics**: Along *low-sharpness directions*, the optimizer progresses steadily, making the primary contribution to loss reduction, albeit at a slow rate.

Inspired by the above picture, a promising approach to accelerating training is as follows: given a base optimizer, increase the LRs along low-sharpness directions while keeping the LR of high-sharpness directions unchanged. This strategy aims to speed up loss reduction without compromising training stability.

Wang et al. (2024) has implemented this idea by adjusting the LR of each parameter based on its sharpness. However, this approach faces two key challenges: 1) it requires frequent diagonal Hessian estimation, which imposes significant computational and memory overhead; 2) sharpness estimates at the individual parameter level can be unreliable.

**The Blockwise LR.** Unlike Wang et al. (2024), we propose adjusting LRs at the block-type level, as our **Principle** (1) reveals a consistent sharpness disparity at this granularity. Specifically, let $\eta_{\mathrm{base}}$ denote the LR for base optimizers such as AdamW, the LR for each block type is then adjusted as follows:

- Norm blocks (the sharpest directions): we still use the base LR, $\eta_{\mathsf{Norm}} = \eta_{\mathrm{base}}$, to keep training stability;

- Other blocks (low-sharpness directions): we adjust the LRs of these blocks by $\eta_\bullet \propto r(\bullet)\eta_{\mathrm{base}}$, where $\bullet \in \{\mathsf{Emb}, \mathsf{QK}, \mathsf{FFN}, \mathsf{VO}\}$, where $r(\bullet)$ denotes the adjusting ratio for the block type $\bullet$.

Naturally, we can set $r(\bullet) \propto \mathcal{S}(\mathsf{Norm})/\mathcal{S}(\bullet)$. However, in practice, we find that manually tuning $r(\bullet)$'s–involving only four hyperparameters–while following the qualitative trend described by **Principle** (1) is more effective. Further details are provided in Section 6.

It is also worth noting that due to its simplicity, Blockwise LR can be seamlessly integrated into modern LLM training frameworks such as Megatron (Shoeybi et al., 2019).

## 6. Experiments

**Models and datasets.** We evaluate our proposed Blockwise LR in the pre-training of decoder-only LLMs across various model types, model sizes, and datasets[1]. Specifically, we

---

[1]The code is available at https://github.com/Wongboo/BlockwiseLearningRate.

consider two widely-used LLMs: **LLaMA** and **GPT-2**; we experiment with model sizes ranging **from 0.12B to 2B** parameters; the datasets includes OpenWebText (Gokaslan & Cohen, 2019) [2], MiniPile (Kaddour, 2023)[3], and Colossal Clean Crawled Corpus (C4) (Raffel et al., 2020)[4], providing a highly diverse text corpus.

**Baselines.** As a baseline, we use the default AdamW optimizer, configured with the hyperparameters $\beta_1 = 0.9, \beta_2 = 0.95$ and weight decay $\lambda = 0.1$. To ensure training stability, gradient clipping is applied with $1.0$. These settings align with the training protocols used in nanoGPT and LLaMA models (Touvron et al., 2023). The LR strategy includes a linear warm-up phase followed by a cosine decay scheduler, capped at `lr_max`. And the terminal LR `lr_min` is set to `lr_max`/20. For each experiment, we *first tune* the `lr_max` to be optimal for AdamW, and the baselines are trained using these optimal `lr_max`'s. Details of the tuned `lr_max` values can be found in Appendix A.1.

**Adjusting ratio tuning and its transferability.** To incorporate the Blockwise LR into AdamW, we simply use the `lr_max` (tuned for vanilla AdamW) for Norm blocks. Then, we **only tuned the four adjusting ratios in a single small-scale experiment** – specifically the pre-training of LLaMA (0.25B) on Minipile – following the rule: $r(\bullet)$ is adjusted according to the trend of $\frac{\mathcal{S}(\mathsf{Norm})}{\mathcal{S}(\bullet)}$, guided by **Principle** (1). The tuned hyperparameters are:

$$r(\mathsf{Emb}) = 10, r(\mathsf{QK}) = 8, r(\mathsf{FFN}) = 6, r(\mathsf{VO}) = 4. \quad (5)$$

Notably, the adjusting ratios are highly robust hyperparameters, as demonstrated in the following ways:

- First, as shown in Figure 10, in the experiments for tuning the adjusting ratios, Blockwise LR demonstrates robustness to these hyperparameters, consistently accelerating pre-training across a range of $r(\bullet)$'s. The configuration in (5) achieves the largest improvements among those tested. Notably, even with suboptimal ratios, Blockwise LR still delivers significant performance gains. Further details are provided in Appendix A.2.

- Second, the configuration in (5), tuned from a single experiment, **transfers perfectly** across all AdamW experiments conducted in this paper. Consequently, **we adopt** (5) **as the default adjusting ratios for all AdamW experiments**. This robustness aligns with the consistency of **Principle** (1), which holds across GPT and LLaMA models, various model sizes, and datasets.

---

[2] An opensource recreation of the WebText corpus, widely used for LLM pre-training such as RoBERTa (Liu et al., 2019) and GPT-2.

[3] A 6GB subset of the deduplicated Pile (825GB) (Gao et al., 2020)

[4] A large-scale public language datasets, widely used for LLM pre-training such as T5 (Raffel et al., 2020)

## 6.1. Main Results

**Main findings.** In Figure 4 and Figures 1(right), we compare the performance of AdamW with Blockwise LR against vanilla AdamW across various settings. Our observations, which consistently hold across all experiments–including both GPT-2 and LLaMA models with sizes ranging from 0.12B to 2B–and datasets including OpenWebText and MiniPile, are as follows:

- Given the same total number of training steps, Blockwise LR enables AdamW to reach a **lower terminal loss** than vanilla AdamW.

- Across different total training steps, AdamW with Blockwise LR achieves a **nearly** $2\times$ **speedup** compared to vanilla AdamW.

An intriguing observation in Figure 4 is that AdamW with Blockwise LR often starts to outperform vanilla AdamW from the mid-to-late stages of training. This behavior resembles the WSD scheduler (Wen et al., 2024; Hu et al., 2024), which typically surpasses cosine or linear decay LR schedulers in the late stage (during the decay phase). Understanding the underlying cause of this phenomenon requires further investigation, which we leave for future work.

**Scaling law is in favor of Blockwise LR.** To further examine scaling behavior, Figure 5 (right) visualizes the scaling laws of AdamW with Blockwise LR versus AdamW during LLaMA pre-training. For MiniPile (left) and OpenWebText (middle), the performance gaps between the two optimizers *get larger as models size grows*. For C4 (right), the performance gap remains stable across model scales, with the corresponding scaling curves remaining nearly parallel. These results suggest that *the gains offered by Blockwise LR may persist at larger model scales*.

**Evaluation on downstream tasks.** Furthermore, as observed in Table 1, the improvement in validation loss transfers to an improvement in downstream task accuracy. Within the same number of pre-training steps, the LLaMA (1.1B) trained with Blockwise LR shows better downstream performance than Adam among all evaluated tasks.

## 6.2. Ablation Studies

In the preceding experiments, Blockwise LR is applied to all major blocks simultaneously. Here, we conduct ablation studies to assess the contribution of each block type individually. Specifically, we pre-train a LLaMA model (0.25B) on OpenWebText focusing on three comparisons: **(i)** applying Blockwise LR exclusively to Emb; **(ii)** applying Blockwise LR to both Emb and FFN; **(iii)** applying Blockwise LR to blocks of all the four types (Emb, FFN, QK, and VO). The adjusting ratios follow Eq. (5) and the results are shown in Table 2.

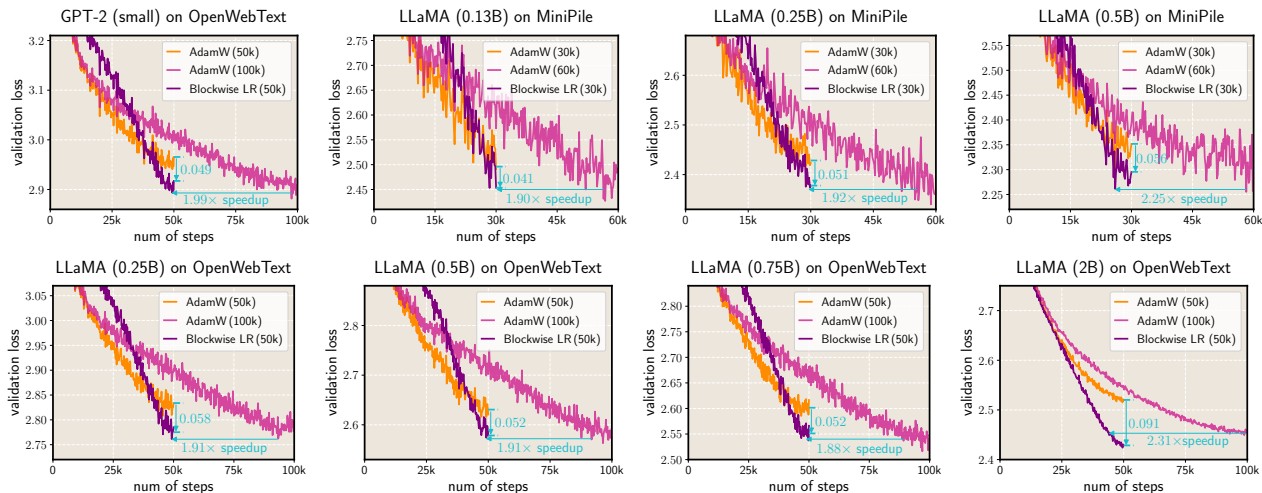

Figure 4: AdamW with Blockwise LR consistently outperforms AdamW in LLM pre-training tasks across different model types, varying model sizes, and datasets.

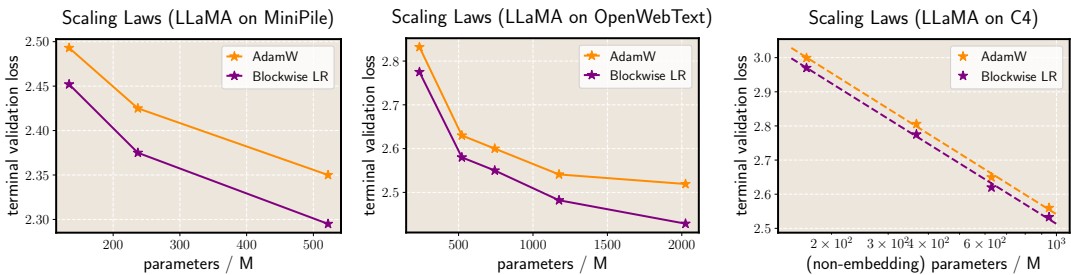

Figure 5: Scaling-law comparison of AdamW with Blockwise LR and AdamW on various datasets for LLaMA models.

Table 1: Evaluation results on downstream tasks (0-shot with lm-evaluation-harness) of LLaMA models (1.1B) pre-trained on OpenWebText using AdamW or Blockwise LR for 50K steps. The best scores in each column are bolded.

| Method | ARC_E | ARC_C | PIQA | HellaSwag | OBQA | WinoGrande | SCIQ |
|--------|-------|-------|------|-----------|------|------------|------|
| AdamW | 52.69 | 22.87 | 68.71 | 36.13 | 19.40 | 55.17 | 77.60 |
| Blockwise LR | **54.29** | **25.34** | **69.53** | **38.00** | **22.60** | **59.83** | **81.60** |

First, the results show that applying Blockwise LR to any block consistently improves performance, supporting the hypothesis that dynamics along low-sharpness directions are crucial for loss reduction. Among all blocks, applying Blockwise LR to FFN yields the largest improvement ($0.043 - 0.016 = 0.027$), likely because FFN blocks comprise the majority of model parameters, offering the greatest potential for optimization gains.

Second, we conduct an additional experiment to assess the impact of increasing the LR for Norm blocks. Specifically, the Norm LR is doubled, while the LR for other blocks remains unchanged from the baseline. As shown in the last row of Table 2, this leads to a deterioration in performance, contrasting with the improvements seen when increasing the LRs for other blocks by far more than double. This result underscores a fundamental difference in the dynamics of

Norm with other blocks.

In summary, these ablation studies further validate the effectiveness of Blockwise LR and confirm the rationale of selecting specific types of blocks for LR amplification, as guided by the sharpness disparity principle.

Table 2: Ablation results for the effectiveness of Blockwise LR in pre-training LLaMA (0.25B) on OpenWebText.

| Blockwise LR | terminal loss (50k steps) |
|--------------|---------------------------|
| w/o | 2.834 |
| Emb | 2.818 (-0.016 ✔) |
| Emb & FFN | 2.791 (-0.043 ✔) |
| Emb & FFN & QK & VO | 2.784 (-0.050 ✔) |
| Norm | 2.837 (+0.003 ✗) |

## 6.3. Integration into Other Optimization Schemes

In practice, there are two popular directions for improving LLM pre-training: acceleration and reducing memory consumption. While Blockwise LR has demonstrated remarkable success in accelerating pre-training, a natural **question** arises: *Can Blockwise LR be combined with memory-efficient optimizers to achieve both faster training and fewer memory consumption?*

**Blockwise LR on Adam-mini.** Without loss of generality, we choose the Adam-mini (Zhang et al., 2024c) optimizer, an Adam variant that reduces memory consumption by approximately $2\times$ compared to AdamW. Here, we conduct experiments to explore whether Blockwise LR can also accelerate Adam-mini. Following Zhang et al. (2024c), we adopt the `lr_max` that tuned for AdamW as the the `lr_max` of Adam-mini. However, since Adam-mini employs SGD within each block, its dynamics differs significantly from AdamW. Consequently, for Adam-mini with Blockwise LR, we re-tune the ratios $r(\bullet)$ for $\bullet \in \{\mathsf{Emb}, \mathsf{QK}, \mathsf{FFN}, \mathsf{VO}\}$. More experimental details are provided in Appendix A.3.

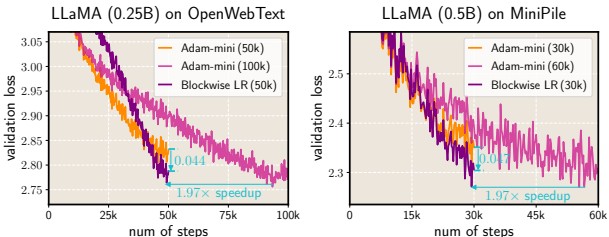

Figure 6: Adam-mini with Blockwise LR outperforms Adam-mini in pre-training tasks.

The results, presented in Figure 6, demonstrate that **Blockwise LR achieves a $2\times$ speedup on Adam-mini**. Since vanilla Adam-mini already achieves a $2\times$ memory saving compared to AdamW while maintaining nearly the same convergence speed, Adam-mini combined with Blockwise LR achieves both a $2\times$ speedup and $2\times$ memory saving compared to vanilla AdamW. We leave more ablation studies with other optimizers for future work.

**Blockwise LR on Lion.** Another memory-efficient optimizer is Lion (Chen et al., 2024), which eliminates second-order moments in AdamW. We conduct experiments to explore whether Blockwise LR can also accelerate Lion. We begin by tuning the `lr_max` for Lion baseline, as detailed in Appendix A.3. For Lion with Blockwise LR, we directly apply the ratios $r(\bullet)$ in Eq. (5) (note that this is originally tuned for AdamW with Blockwise LR). The results, presented in Figure 7 (left), demonstrate that *Blockwise LR yields a $2\times$ speedup on Lion*.

Additionally, we evaluate the evolution of the *average sharpness across different blocks* when trained using *Lion optimizer*. As shown in Figure 11 in Appendix A.3, the results

*closely resemble those* in Figure 3(b), which uses AdamW. Our Principle (Eq. (1)) emerges during the initial phase, and persists throughout the subsequent training process.

**Blockwise LR with wsd scheduler.** The preceding experiments employ the cosine decayed LR scheduler. In this section, we evaluate Blockwise LR under an alternative and increasingly popular scheduler: warmup-stable-decay (wsd) (Hu et al., 2024), which includes a linear warm-up LR to peak `lr_max`, followed by a stable phase where LR remains at `lr_max`, and then a linear decay to `lr_min`. We extend our experiments to incorporate the WSD scheduler. Experimental details are provided in Appendix A.3. As shown in Figure 7 (right), *Blockwise LR still achieves a $2\times$ speedup when used with the wsd scheduler*.

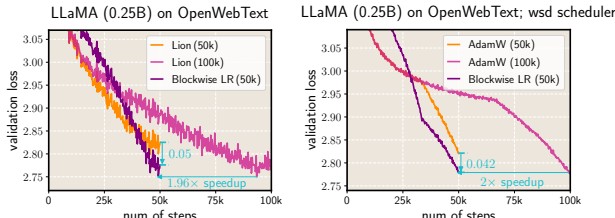

Figure 7: In pre-training tasks, (left) Lion with Blockwise LR outperforms Lion; (right) when using wsd LR scheduler, AdamW with Blockwise LR outperforms AdamW.

These experiments demonstrate that Blockwise LR is not limited to accelerating AdamW but can also be effectively combined with other optimizers such as Adam-mini and Lion, and LR scheduler such as wsd, while preserving their unique advantages. This finding paves the way for future research exploring the integration of Blockwise LR with other optimization algorithms.

## 7. Conclusion and Outlook

In this paper, we uncovered a sharpness disparity principle among different types of blocks in transformers, as formalized in Eq. (1). Notably, this blockwise sharpness disparity persists throughout the entire training process, except during the initial few steps. Building on this discovery, we proposed a novel Blockwise LR adjustment principle, which effectively accelerates base optimizers such as AdamW and Adam-mini in LLM pre-training tasks.

**Future works.** It would be valuable to investigate the applicability of our Blockwise LR to non-LLM tasks, such as computer vision, and its compatibility with other optimizers, such as Muon (Keller et al., 2024) and other alloy-like architectures such as Mamba (Gu & Dao, 2023). Furthermore, our findings open up opportunities to develop other block-adaptive optimization strategies, such as blockwise weight decay and gradient clipping, which could further enhance training efficiency and performance.

## Acknowledgements

Lei Wu was supported by the National Key R&D Program of China (No. 2022YFA1008200) and National Natural Science Foundation of China (No. 12288101). Mingze Wang was supported by Young Scientists (PhD) Fund of the National Natural Science Foundation of China (No. 124B2028). Junchi Yan and Zhanpeng Zhou were partly supported by NSFC (72342023).

## Impact Statement

This paper contributes to advancing the field of deep learning, with a focus on understanding and improving the pretraining of LLMs. While our work has the potential to impact society in various ways, we do not identify any specific societal consequences that require particular emphasis at this time.

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

## A. Experimental Details

**Models.** We utilize two popular classes of LLM models for our pre-training experiments:

- **GPT-2.** We use GPT-2 (small) model (Radford et al., 2019), implemented via the nanoGPT code base (Karpathy, 2022). Following nanoGPT, the model employs Gaussian Error Linear Unit (GELU) activations and standard Layer Normalization (LayerNorm). Detailed model configurations are provided in Table 3.

- **LLaMA.** LLaMA (Touvron et al., 2023) is another popular decoder-only Transformer architecture, incorporating Rotary Positional Encoding (RoPE) (Su et al., 2024), Swish-Gated Linear Unit (SwiGLU), and Root mean square layer normalization (RMSNorm). We pre-train LLaMA models of sizes ranging from 0.13B to 2B parameters. For implementation, for the 1.1B model configuration, we follow TinyLlama (Zhang et al., 2024a), which employs grouped-query attention (Ainslie et al., 2023); for other model sizes, we utilize the LLaMA code from HuggingFace Transformers Library (Wolf et al., 2020). Additional model configurations are detailed in Table 3 and 4.

**Datasets.** Models are pre-trained on the following datasets:

- **OpenWebText** (Gokaslan & Cohen, 2019). It is an opensource recreation of the WebText corpus, is extensively utilized for LLM pre-training such as RoBERTa (Liu et al., 2019) and GPT-2.

- **MiniPile.** (Kaddour, 2023). It is a 6GB subset of the deduplicated Pile (825GB) (Gao et al., 2020) presents a highly diverse text corpus. Given its diversity, training on minipile poses challenges and potential instabilities.

- **Colossal Clean Crawled Corpus (C4)** (Raffel et al., 2020). It is a large-scale public language dataset, widely used for LLM pre-training such as T5 (Raffel et al., 2020), and prior pre-training studies (Zhao et al., 2024a;b).

All experiments are conducted on 4 A800/H800 80G GPUs.

### A.1. Training Configurations for AdamW Baselines

Table 3: Model configurations and optimally-tuned peak learning rates on OpenWebText and MiniPile.

| Acronym | Size | $d_{\mathrm{model}}$ | $d_{\mathrm{FF}}$ | n_head | depth | lr_max on OpenWebText | lr_max on MiniPile |
|---|---|---|---|---|---|---|---|
| GPT-2 (small) | 124M | 768 | 3072 | 12 | 12 | 6e-4 | 6e-4 |
| LLaMA (0.13B) | 134M | 768 | 3072 | 12 | 6 | – | 1.2e-3 |
| LLaMA (0.25B) | 237M | 1024 | 4096 | 16 | 8 | 8e-4 | 7.5e-4 |
| LLaMA (0.5B) | 522M | 1280 | 5120 | 20 | 15 | 8e-4 | 4.5e-4 |
| LLaMA (0.75B) | 743M | 1664 | 6656 | 26 | 13 | 6e-4 | – |
| LLaMA (1.1B) | 1175M | 2048 | 5632 | 32 | 22 | 4e-4 | – |
| LLaMA (2B) | 2025M | 2048 | 8192 | 32 | 22 | 2e-4 | – |

Table 4: Model configurations and optimally-tuned peak learning rates on C4.

| Acronym | Size | $d_{\mathrm{model}}$ | $d_{\mathrm{FF}}$ | n_head | depth | lr_max |
|---|---|---|---|---|---|---|
| LLaMA (66M) | 66M | 512 | 2048 | 8 | 8 | 1e-3 |
| LLaMA (0.2B) | 200M | 768 | 3072 | 16 | 8 | 1e-3 |
| LLaMA (0.4B) | 400M | 1280 | 5120 | 16 | 12 | 6e-4 |
| LLaMA (1B) | 1004M | 1600 | 6400 | 25 | 22 | 3e-4 |

As a baseline optimizer, we use the default AdamW for LLM pre-training, configured with the hyperparameters $\beta_1 = 0.9, \beta_2 = 0.95$ and weight decay $\lambda = 0.1$. To ensure training stability, gradient clipping is applied by norm with threshold 1.0. These settings align with the training protocols used in nanoGPT and LLaMA models (Touvron et al., 2023). The default LR strategy integrates a linear warm-up phase, followed by a cosine decay scheduler with the peak learning rate lr_max and the final learning rate lr_min=lr_max/20. Additionally,

- **OpenWebText pre-training.** The (max) sequence length is set to 1024, and the batch size is set to 480, following nanoGPT and Liu et al. (2024b). The total training duration is 50,000 or 100,000 steps, including 1,000 warm-up steps. The grid search for `lr_max` is performed over $\{$`2e-4, 4e-4, 6e-4, 8e-4, 1e-3`$\}$. Optimal learning rates for each model are detailed in Table 3.

- **MiniPile pre-training.** The (max) sequence length is set to 512, and the batch size is set to 300, following Wang et al. (2024). The total training duration is 30,000 or 60,000 steps, including 600 warm-up steps. The grid search for `lr_max` is performed over $\{$`3e-4, 4.5e-4, 6e-4, 7.5e-4, 9e-4, 1.2e-3, 1.5e-3`$\}$. Optimal learning rates for each model are detailed in Table 3.

- **C4 pre-training** We follow the setup of Zhao et al. (2024a;b), using a sequence length of 256 and batch size of 512. Following the Chinchilla scaling law (Hoffmann et al., 2022), the total number of training tokens is set to be approximately 20 times the number of model parameters. The training includes 1,000 warm-up steps. The grid search for `lr_max` is performed over $\{$`1e-4, 2e-4, 3e-4, 6e-4, 1e-3, 1.5e-3`$\}$. Optimal learning rates for each model are detailed in Tables 4. We use the T5 tokenizer, with the vocabulary size 32100.

**Baselines**: models are pre-trained using AdamW with the respective tuned `lr_max` for each dataset and model configuration.

**Related Experiments.**

- **Blockwise LR Experiments.** The baseline results in **Figure 4**, **Figure 1 (right)**, and **Table 2 (the w/o line)** are trained following the configurations above.

- **Sharpness Principle Experiments.** Models for **Figure 1 (left)**, **Figure 2**, **Figure 3**, are trained using the baseline configurations for GPT-2 (small) or LLaMA (0.25B) on OpenWebText, with a total training duration 50,000 steps. In these experiments, the sharpness is estimated using $h(\theta)$ in Eq. (3), with $B$ set to 1024. The sharpness distributions and average sharpness values for different blocks (•) are calculated on a logarithmic scale, i.e., $\log h(\theta[\bullet])$.

  Additionally, the experiment in Figure 9 employs the same model and sharpness estimator.

- **Theoretical Analysis Support.** To support our theoretical insights in Section 4.2, **Figure 8** shows the evaluation of the parameter norms across different blocks during training. The model used is LlaMa (0.25B), trained on OpenWebText. The model is LLaMA (0.25B), trained on OpenWebText following the baseline configurations.

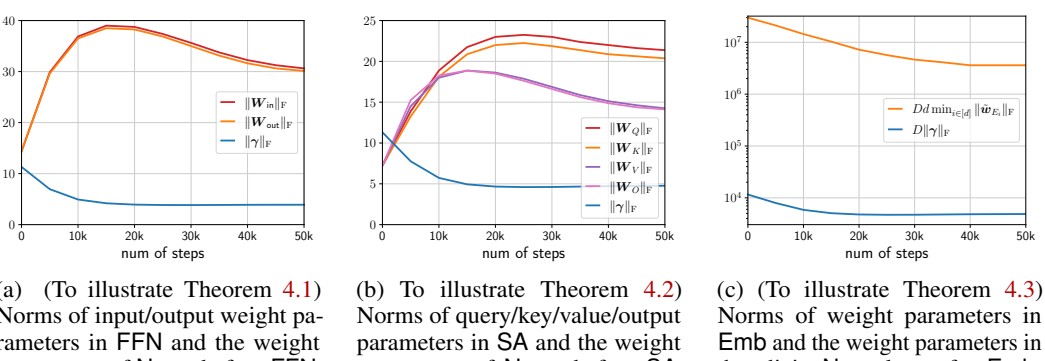

(a) (To illustrate Theorem 4.1) Norms of input/output weight parameters in FFN and the weight parameters of Norm before FFN, averaged by the number of layers.

(b) To illustrate Theorem 4.2) Norms of query/key/value/output parameters in SA and the weight parameters of Norm before SA, averaged by the number of layers.

(c) (To illustrate Theorem 4.3) Norms of weight parameters in Emb and the weight parameters in the adjoint Norm layer after Emb.

Figure 8: Evolution of parameter norms across different blocks during pre-training LLaMA (0.25B) on OpenWebText.

## A.2. Experimental Details for Blockwise LR on AdamW

**Switching Time.** The principle of blockwise sharpness heterogeneity emerges clearly after the initial training phase, as shown in Figure 3. To leverage this principle, in our experiments of AdamW using Blockwise LR, we **switch** from standard AdamW to AdamW with Blockwise LR **at the end of LR warmup phase**.

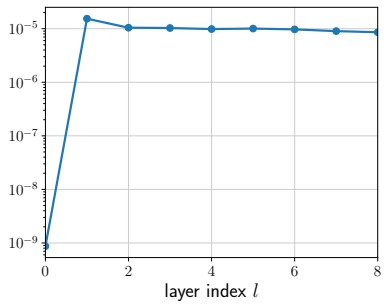 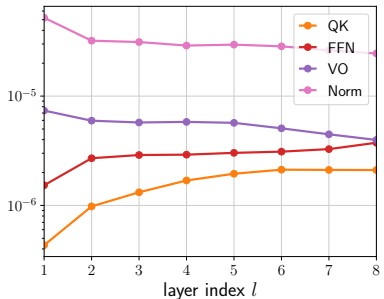

(a) Average sharpness across different layers. Layer $0$ corresponds to the Emb layer. Layers $1, \cdots, 8$ correspond to the SA-FFN layers.

(b) Average sharpness of the blocks ($\bullet \in \{\mathsf{QK}, \mathsf{FFN}, \mathsf{VO}, \mathsf{Norm}\}$) across different layers ($l = 1, \cdots, 8$).

Figure 9: In a pre-trained LLaMA (0.25B) (with $L = 8$ layer), there is no clear disparity for the average sharpness across the **layers**. This is in stark contrast to our our sharpness disparity **Principle** (1) across the **blocks**.

**Experiments in Figure 4.** We adopt the adjusting ratios (5) as the default adjusting ratios for all experiments of AdamW with Blockwise LR.

**Experiment on Hyper-parameter Tuning.** We **only** tune the four adjusting ratios $r(\bullet)$ ($\bullet \in \{\mathsf{Emb}, \mathsf{QK}, \mathsf{VO}, \mathsf{FFN}\}$) in a single small-scale experiment: pre-training LLaMA (0.25B) on Minipile. Specifically, we compare the results under the following configurations of ratios:

$$r(\mathsf{Emb}) = 6, r(\mathsf{QK}) = 4, r(\mathsf{FFN}) = 3, r(\mathsf{VO}) = 2;$$
$$r(\mathsf{Emb}) = 8, r(\mathsf{QK}) = 6, r(\mathsf{FFN}) = 4, r(\mathsf{VO}) = 3;$$
$$r(\mathsf{Emb}) = 10, r(\mathsf{QK}) = 8, r(\mathsf{FFN}) = 6, r(\mathsf{VO}) = 4.$$

The results for the tuning experiments are presented in **Figure 10**. One can see that the configuration $r(\mathsf{Emb}) = 10, r(\mathsf{QK}) = 8, r(\mathsf{FFN}) = 6, r(\mathsf{VO}) = 4$ (Eq. (5)) achieves the largest improvement in terminal loss. Additionally, Blockwise LR demonstrates robustness to these ratios, consistently accelerating pre-training across all tested configurations.

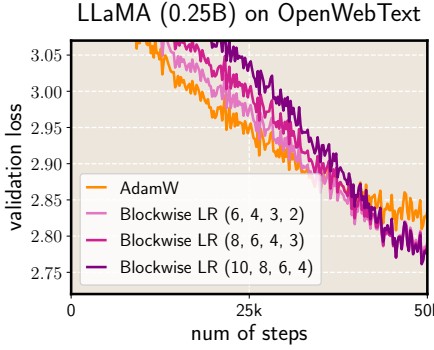

Figure 10: Pre-training LLaMA (0.25B) on Minipile using AdamW with Blockwise LR across three configurations of adjusting ratios.

**Experiments in Table 2.** We pre-train LLaMA (0.25B) on OpenWebText with a focusing on the three comparisons: (i) applying Blockwise LR exclusively to Emb; (ii) applying Blockwise LR to both Emb and FFN; (iii) applying Blockwise LR to blocks of all the four types (Emb, FFN, QK, and VO). The adjusting ratios are maintained as per the tuned in Eq. (5).

### A.3. Experimental details for Adam-mini, Lion, and wsd

**Experiments for Adam-mini.**

- **Baseline.** In the baseline experiments in Figure 6, following Zhang et al. (2024c), we adopt the same peak learning rate lr_max tuned for AdamW as the lr_max of Adam-mini.

- **Hyperparameter tuning.** Since Adam-mini uses SGD within each blocks, its dynamics **differs significantly** from those of AdamW. Thus, for Adam-mini with Blockwise LR, we re-tune the ratios $r(\bullet) \in \{1, 2, 4\}$ for $\bullet \in \{\mathsf{Emb}, \mathsf{QK}, \mathsf{FFN}, \mathsf{VO}\}$. The tuned ratios are $r(\mathsf{Emb}) = 4, r(\mathsf{QK}) = 1, r(\mathsf{FFN}) = 4, r(\mathsf{VO}) = 4$, which are used in the experiments in **Figure 6**. Note that these ratios do not satisfy $r(\bullet) \propto \frac{\mathcal{S}(\mathsf{Norm})}{\mathcal{S}(\bullet)}$. This discrepancy may stem from the unique dynamics of Adam-mini, particularly its SGD-like behavior within blocks. We leave further investigation for future work.

**Experiments for Lion** In the baseline experiments in Figure 7 (left), following Chen et al. (2024), we search for the optimal maximum learning rate lr_max for Lion within the set $\{1/3, 1/5, 1/10\}$ of the lr_max values tuned for AdamW, using corresponding weight decay values of $\{0.3, 0.5, 1.0\}$. Ultimately, we adopt 1/5 of the AdamW-tuned lr_max with a weight decay of 0.5. For Lion with Blockwise LR, we directly apply the ratios in Eq. (5) (note that this is originally tuned for AdamW with Blockwise LR).

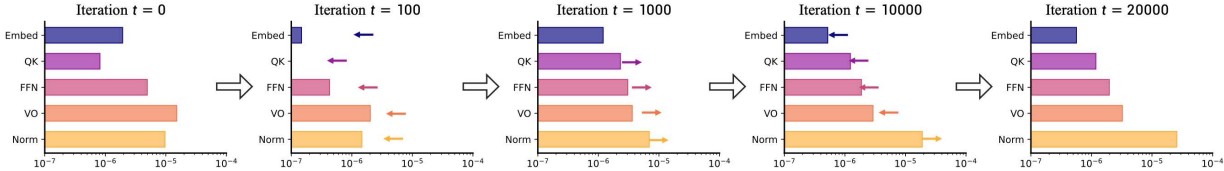

Figure 11: Evolution of the average sharpness across different blocks during pre-training LLaMA (0.25B) on OpenWebText using the **Lion optimizer**. The total training duration is 50k steps. The results closely *resemble those* in our Figure 3(b), which uses AdamW. Our Principle (Eq. (1)) emerges during the initial phase (from iteration 0 to iteration 1k), which accounts for only approximately 2% of the total steps, and persists throughout the subsequent training process.

**Experiments for wsd sheduler.** In the experiments in Figure 7 (right), we use a linear warm-up LR to peak lr_max, followed by a stable phase where LR remains at lr_max (up to 66.7% of the total training steps), and then a linear decay to 0. The lr_max is identical to that used in the cosine decay scheduler, as reported in Table 3.

# B. Proofs in Section 4

### B.1. Proof of Theorem 4.1

We focus on the transformation from $\boldsymbol{X}^{(l-1)}$ to $\boldsymbol{X}^{(l-1/2)}$:

$$\boldsymbol{X}^{(l)} = \boldsymbol{X}^{(l-1/2)} + \mathsf{FFN}^{(l)}\left(\mathsf{Norm}^l\left(\boldsymbol{X}^{(l-1/2)}; \boldsymbol{\gamma}^{(l)}\right); \boldsymbol{W}_1^{(l)}, \boldsymbol{W}_2^{(l)}\right).$$

From the chain rule, it follows that:

$$\frac{\partial \mathcal{Q}}{\partial \boldsymbol{W}_\bullet^{(l)}} = \frac{\partial \mathcal{Q}}{\partial \boldsymbol{X}^{(l)}} \frac{\partial \boldsymbol{X}^{(l)}}{\partial \boldsymbol{W}_\bullet^{(l)}}, \quad \bullet \in \{1, 2\};$$

$$\frac{\partial \mathcal{Q}}{\partial \boldsymbol{\gamma}^{(l)}} = \frac{\partial \mathcal{Q}}{\partial \boldsymbol{X}^{(l)}} \frac{\partial \boldsymbol{X}^{(l)}}{\partial \boldsymbol{\gamma}^{(l)}}.$$

Thus, it suffices to compute $\frac{\partial \boldsymbol{X}^{(l)}}{\partial \boldsymbol{W}_\bullet^{(l)}}$ and $\frac{\partial \boldsymbol{X}^{(l)}}{\partial \boldsymbol{\gamma}^{(l)}}$. For simplicity, we define:

$$\boldsymbol{X} := \boldsymbol{X}^{(l-1/2)}, \quad \boldsymbol{X}_{\mathrm{std}} = \frac{\boldsymbol{X} - \mathbb{E}_r[\boldsymbol{X}]}{\sqrt{\mathbb{V}_r[\boldsymbol{X}]}}, \quad \boldsymbol{X}_{\mathsf{Norm}} := \mathsf{Norm}(\boldsymbol{X}; \boldsymbol{\gamma}) = \boldsymbol{X}_{\mathrm{std}} \odot (\mathbf{1}_{n \times 1} \otimes \boldsymbol{\gamma}),$$

$$\mathsf{M} := \boldsymbol{X}_{\mathsf{Norm}} \boldsymbol{W}_1, \quad \mathsf{A} := \sigma(\mathsf{M}), \quad \mathsf{F} := \mathsf{A}\boldsymbol{W}_2, \quad \boldsymbol{Y} := \boldsymbol{X}^{(l)} = \boldsymbol{X} + \mathsf{F},$$

where $\sigma(\cdot)$ represents the ReLU or Leacky ReLU activation function. We now compute $\frac{\partial \boldsymbol{Y}}{\partial \boldsymbol{W}_\bullet}$ and $\frac{\partial \boldsymbol{Y}}{\partial \gamma}$.

It is straightforward that:

$$\frac{\partial \boldsymbol{Y}}{\partial \boldsymbol{W}_1} = \frac{\partial \mathsf{F}}{\partial \boldsymbol{W}_1} = \frac{\partial \mathsf{F}}{\partial \mathsf{A}}\frac{\partial \mathsf{A}}{\partial \mathsf{M}}\frac{\partial \mathsf{M}}{\partial \boldsymbol{W}_1} = \left(\boldsymbol{I}_n \otimes \boldsymbol{W}_2^\top\right)\frac{\partial \mathsf{A}}{\partial \mathsf{M}}\left(\boldsymbol{X}_{\mathsf{Norm}} \otimes \boldsymbol{I}_M\right);$$

$$\begin{aligned}\frac{\partial \boldsymbol{Y}}{\partial \gamma} &= \frac{\partial \mathsf{F}}{\partial \boldsymbol{X}_{\mathsf{Norm}}}\frac{\partial \boldsymbol{X}_{\mathsf{Norm}}}{\partial \gamma} = \frac{\partial \mathsf{F}}{\partial \mathsf{A}}\frac{\partial \mathsf{A}}{\partial \mathsf{M}}\frac{\partial \mathsf{M}}{\partial \boldsymbol{X}_{\mathsf{Norm}}}\frac{\partial \boldsymbol{X}_{\mathsf{Norm}}}{\partial \gamma} \\ &= \left(\boldsymbol{I}_n \otimes \boldsymbol{W}_2^\top\right)\frac{\partial \mathsf{A}}{\partial \mathsf{M}}\left(\boldsymbol{I}_n \otimes \boldsymbol{W}_1^\top\right)\left(\mathrm{diag}\big(\mathrm{vec}(\boldsymbol{X}_{\mathrm{std}})\big)\left(\boldsymbol{1}_{n\times 1} \otimes \boldsymbol{I}_D\right)\right).\end{aligned}$$

For the (Leaky) ReLU, it holds that $\sigma(z) = z\sigma'(z)$. Thus, for $\frac{\partial \boldsymbol{Y}}{\partial \boldsymbol{W}_2}$, we have:

$$\frac{\partial \boldsymbol{Y}}{\partial \boldsymbol{W}_2} = \frac{\partial \mathsf{F}}{\partial \boldsymbol{W}_2} = \mathsf{A} \otimes \boldsymbol{I}_D = \left(\boldsymbol{X}_{\mathsf{Norm}}\boldsymbol{W}_1 \odot \frac{\partial \mathsf{A}}{\partial \mathsf{M}}\right) \otimes \boldsymbol{I}_D.$$

Now we derive the upper bounds. First, notice that:

$$\|\boldsymbol{X}_{\mathrm{std}}\|_{\mathrm{F}} = \left(\sum_{i=1}^n \left(\frac{\boldsymbol{X}_{i,:} - \mathbb{E}[\boldsymbol{X}_{i,:}]}{\sqrt{\mathbb{V}[\boldsymbol{X}_{i,:}]}}\right)^2\right)^{1/2} = \left(\sum_{i=1}^n D\right)^{1/2} = \sqrt{nD};$$

$$\|\boldsymbol{X}_{\mathsf{Norm}}\|_{\mathrm{F}} = \|\boldsymbol{X}_{\mathrm{std}} \odot (\boldsymbol{1}_{n\times 1} \otimes \gamma)\|_{\mathrm{F}} \leq \|\boldsymbol{X}_{\mathrm{std}}\|_{\mathrm{F}}\|\boldsymbol{1}_{n\times 1} \otimes \gamma\|_{\mathrm{F}} \leq \sqrt{nD}\|\boldsymbol{1}_{n\times 1}\|_{\mathrm{F}}\|\gamma\|_{\mathrm{F}} \leq n\sqrt{D}\|\gamma\|_{\mathrm{F}}.$$

Consequently, we have the following estimates:

$$\begin{aligned}\left\|\frac{\partial \mathcal{Q}}{\partial \boldsymbol{W}_1}\right\|_{\mathrm{F}} &\leq \left\|\frac{\partial \mathcal{Q}}{\partial \boldsymbol{Y}}\right\|_{\mathrm{F}}\left\|\frac{\partial \boldsymbol{Y}}{\partial \boldsymbol{W}_1}\right\|_{\mathrm{F}} = \left\|\frac{\partial \mathcal{Q}}{\partial \boldsymbol{Y}}\right\|_{\mathrm{F}}\left\|\left(\boldsymbol{I}_n \otimes \boldsymbol{W}_2^\top\right)\frac{\partial \mathsf{A}}{\partial \mathsf{M}}\left(\boldsymbol{X}_{\mathsf{Norm}} \otimes \boldsymbol{I}_M\right)\right\|_{\mathrm{F}} \\ &\leq \left\|\frac{\partial \mathcal{Q}}{\partial \boldsymbol{Y}}\right\|_{\mathrm{F}}\left\|\frac{\partial \mathsf{A}}{\partial \mathsf{M}}\right\|_{\mathrm{F}}\|\boldsymbol{I}_n \otimes \boldsymbol{W}_2^\top\|_2\|\boldsymbol{X}_{\mathsf{Norm}} \otimes \boldsymbol{I}_M\|_2 \leq \left\|\frac{\partial \mathcal{Q}}{\partial \boldsymbol{Y}}\right\|_{\mathrm{F}}\left\|\frac{\partial \mathsf{A}}{\partial \mathsf{M}}\right\|_{\mathrm{F}}\|\boldsymbol{I}_n\|_2\|\boldsymbol{I}_M\|_2\|\boldsymbol{W}_2^\top\|_{\mathrm{F}}\|\boldsymbol{X}_{\mathsf{Norm}}\|_{\mathrm{F}} \\ &\leq \left\|\frac{\partial \mathcal{Q}}{\partial \boldsymbol{Y}}\right\|_{\mathrm{F}}\left\|\frac{\partial \mathsf{A}}{\partial \mathsf{M}}\right\|_{\mathrm{F}}\|\boldsymbol{W}_2\|_{\mathrm{F}}\|\boldsymbol{X}_{\mathsf{Norm}}\|_{\mathrm{F}} \leq n\sqrt{D}\left\|\frac{\partial \mathcal{Q}}{\partial \boldsymbol{Y}}\right\|_{\mathrm{F}}\left\|\frac{\partial \mathsf{A}}{\partial \mathsf{M}}\right\|_{\mathrm{F}}\|\boldsymbol{W}_2\|_{\mathrm{F}}\|\gamma\|_{\mathrm{F}};\end{aligned}$$

$$\begin{aligned}\left\|\frac{\partial \mathcal{Q}}{\partial \boldsymbol{W}_2}\right\|_{\mathrm{F}} &\leq \left\|\frac{\partial \mathcal{Q}}{\partial \boldsymbol{Y}}\right\|_{\mathrm{F}}\left\|\frac{\partial \boldsymbol{Y}}{\partial \boldsymbol{W}_2}\right\|_{\mathrm{F}} = \left\|\frac{\partial \mathcal{Q}}{\partial \boldsymbol{Y}}\right\|_{\mathrm{F}}\left\|\left(\boldsymbol{X}_{\mathsf{Norm}}\boldsymbol{W}_1 \odot \frac{\partial \mathsf{A}}{\partial \mathsf{M}}\right) \otimes \boldsymbol{I}_D\right\|_{\mathrm{F}} \\ &\leq \left\|\frac{\partial \mathcal{Q}}{\partial \boldsymbol{Y}}\right\|_{\mathrm{F}}\left\|\left(\boldsymbol{X}_{\mathsf{Norm}}\boldsymbol{W}_1 \odot \frac{\partial \mathsf{A}}{\partial \mathsf{M}}\right)\right\|_{\mathrm{F}}\|\boldsymbol{I}_D\|_2 \leq \left\|\frac{\partial \mathcal{Q}}{\partial \boldsymbol{Y}}\right\|_{\mathrm{F}}\left\|\frac{\partial \mathsf{A}}{\partial \mathsf{M}}\right\|_{\mathrm{F}}\|\boldsymbol{X}_{\mathsf{Norm}}\boldsymbol{W}_1\|_{\mathrm{F}} \\ &\leq \left\|\frac{\partial \mathcal{Q}}{\partial \boldsymbol{Y}}\right\|_{\mathrm{F}}\left\|\frac{\partial \mathsf{A}}{\partial \mathsf{M}}\right\|_{\mathrm{F}}\|\boldsymbol{W}_1\|_{\mathrm{F}}\|\boldsymbol{X}_{\mathsf{Norm}}\|_{\mathrm{F}} \leq n\sqrt{D}\left\|\frac{\partial \mathcal{Q}}{\partial \boldsymbol{Y}}\right\|_{\mathrm{F}}\left\|\frac{\partial \mathsf{A}}{\partial \mathsf{M}}\right\|_{\mathrm{F}}\|\boldsymbol{W}_1\|_{\mathrm{F}}\|\gamma\|_{\mathrm{F}};\end{aligned}$$

$$\begin{aligned}\left\|\frac{\partial \mathcal{Q}}{\partial \gamma}\right\|_{\mathrm{F}} &\leq \left\|\frac{\partial \mathcal{Q}}{\partial \boldsymbol{Y}}\right\|_{\mathrm{F}}\left\|\frac{\partial \boldsymbol{Y}}{\partial \gamma}\right\|_{\mathrm{F}} \\ &= \left\|\frac{\partial \mathcal{Q}}{\partial \boldsymbol{Y}}\right\|_{\mathrm{F}}\left\|\left(\boldsymbol{I}_n \otimes \boldsymbol{W}_2^\top\right)\frac{\partial \mathsf{A}}{\partial \mathsf{M}}\left(\boldsymbol{I}_n \otimes \boldsymbol{W}_1^\top\right)\left(\mathrm{diag}\big(\mathrm{vec}(\boldsymbol{X}_{\mathrm{std}})\big)\left(\boldsymbol{1}_{n\times 1} \otimes \boldsymbol{I}_D\right)\right)\right\|_{\mathrm{F}} \\ &\leq \left\|\frac{\partial \mathcal{Q}}{\partial \boldsymbol{Y}}\right\|_{\mathrm{F}}\left\|\frac{\partial \mathsf{A}}{\partial \mathsf{M}}\right\|_{\mathrm{F}}\|\boldsymbol{I}_n \otimes \boldsymbol{W}_2^\top\|_{\mathrm{F}}\|\boldsymbol{I}_n \otimes \boldsymbol{W}_1^\top\|_{\mathrm{F}}\|\mathrm{diag}\big(\mathrm{vec}(\boldsymbol{X}_{\mathrm{std}})\big)\left(\boldsymbol{1}_{n\times 1} \otimes \boldsymbol{I}_D\right)\|_{\mathrm{F}} \\ &\leq \left\|\frac{\partial \mathcal{Q}}{\partial \boldsymbol{Y}}\right\|_{\mathrm{F}}\left\|\frac{\partial \mathsf{A}}{\partial \mathsf{M}}\right\|_{\mathrm{F}}\|\boldsymbol{I}_n\|_2\|\boldsymbol{W}_2\|_{\mathrm{F}}\|\boldsymbol{I}_n\|_2\|\boldsymbol{W}_1\|_{\mathrm{F}}\|\mathrm{diag}\big(\mathrm{vec}(\boldsymbol{X}_{\mathrm{std}})\big)\|_{\mathrm{F}}\|\boldsymbol{1}_{n\times 1} \otimes \boldsymbol{I}_D\|_{\mathrm{F}}\end{aligned}$$

$$\leq \left\| \frac{\partial \mathcal{Q}}{\partial \boldsymbol{Y}} \right\|_{\mathrm{F}} \left\| \frac{\partial \mathsf{A}}{\partial \mathsf{M}} \right\|_{\mathrm{F}} \|\boldsymbol{W}_1\|_{\mathrm{F}} \|\boldsymbol{W}_2\|_{\mathrm{F}} \|\boldsymbol{X}_{\mathrm{std}}\|_{\mathrm{F}} \|\boldsymbol{1}_{n \times 1}\|_{\mathrm{F}} \|\boldsymbol{I}_D\|_2$$

$$\leq n\sqrt{D} \left\| \frac{\partial \mathcal{Q}}{\partial \boldsymbol{Y}} \right\|_{\mathrm{F}} \left\| \frac{\partial \mathsf{A}}{\partial \mathsf{M}} \right\|_{\mathrm{F}} \|\boldsymbol{W}_1\|_{\mathrm{F}} \|\boldsymbol{W}_2\|_{\mathrm{F}}.$$

Thus, if we define

$$\Psi := n\sqrt{D} \left\| \frac{\partial \mathcal{Q}}{\partial \boldsymbol{Y}} \right\|_{\mathrm{F}} \left\| \frac{\partial \mathsf{A}}{\partial \mathsf{M}} \right\|_{\mathrm{F}} \|\boldsymbol{W}_1\|_{\mathrm{F}} \|\boldsymbol{W}_2\|_{\mathrm{F}} \|\boldsymbol{\gamma}\|_{\mathrm{F}},$$

then it holds that:

$$\left\| \frac{\partial \mathcal{Q}}{\partial \boldsymbol{W}_1} \right\|_{\mathrm{F}} \leq \frac{\Psi}{\|\boldsymbol{W}_1\|_{\mathrm{F}}}; \quad \left\| \frac{\partial \mathcal{Q}}{\partial \boldsymbol{W}_2} \right\|_{\mathrm{F}} \leq \frac{\Psi}{\|\boldsymbol{W}_2\|_{\mathrm{F}}}; \quad \left\| \frac{\partial \mathcal{Q}}{\partial \boldsymbol{\gamma}} \right\|_{\mathrm{F}} \leq \frac{\Psi}{\|\boldsymbol{\gamma}\|_{\mathrm{F}}}$$

Therefore,

$$\mathcal{S}(\boldsymbol{W}_\bullet) = \frac{1}{\#(\boldsymbol{W}_\bullet)} \left\| \frac{\partial \mathcal{Q}}{\partial \boldsymbol{W}_\bullet} \right\|_{\mathrm{F}}^2 = \mathcal{O}\left( \frac{\Psi^2}{D^2 \|\boldsymbol{W}_\bullet\|_{\mathrm{F}}^2} \right), \quad \bullet \in \{1, 2\};$$

$$\mathcal{S}(\boldsymbol{\gamma}) = \frac{1}{\#(\boldsymbol{\gamma})} \left\| \frac{\partial \mathcal{Q}}{\partial \boldsymbol{\gamma}} \right\|_{\mathrm{F}}^2 = \mathcal{O}\left( \frac{\Psi^2}{D \|\boldsymbol{\gamma}\|_{\mathrm{F}}^2} \right).$$

### B.2. Proof of Theorem 4.2

We focus on the transformation from $\boldsymbol{X}^{(l-1)}$ to $\boldsymbol{X}^{(l-1/2)}$:

$$\boldsymbol{X}^{(l-1/2)} = \boldsymbol{X}^{(l-1)} + \mathsf{SA}^{(l)}\left( \mathsf{Norm}^{(l-1/2)}\left( \boldsymbol{X}^{(l-1)}; \boldsymbol{\gamma}^{(l-1/2)} \right); \boldsymbol{W}_K^{(l)}, \boldsymbol{W}_Q^{(l)}, \boldsymbol{W}_V^{(l)}, \boldsymbol{W}_O^{(l)} \right).$$

From the chain rule, it follows that:

$$\frac{\partial \mathcal{Q}}{\partial \boldsymbol{W}_\bullet^{(l)}} = \frac{\partial \mathcal{Q}}{\partial \boldsymbol{X}^{(l-1/2)}} \frac{\partial \boldsymbol{X}^{(l-1/2)}}{\partial \boldsymbol{W}_\bullet^{(l)}}, \quad \bullet \in \{K, Q, V, O\};$$

$$\frac{\partial \mathcal{Q}}{\partial \boldsymbol{\gamma}^{(l-1/2)}} = \frac{\partial \mathcal{Q}}{\partial \boldsymbol{X}^{(l-1/2)}} \frac{\partial \boldsymbol{X}^{(l-1/2)}}{\partial \boldsymbol{\gamma}^{(l-1/2)}}.$$

Thus, it suffices to compute $\frac{\partial \boldsymbol{X}^{(l-1/2)}}{\partial \boldsymbol{W}_\bullet^{(l-1/2)}}$ and $\frac{\partial \boldsymbol{X}^{(l-1/2)}}{\partial \boldsymbol{\gamma}^{(l-1/2)}}$. For simplicity, we define:

$$\boldsymbol{X} := \boldsymbol{X}^{(l-1)}, \quad \boldsymbol{X}_{\mathrm{std}} = \frac{\boldsymbol{X} - \mathbb{E}_r[\boldsymbol{X}]}{\sqrt{\mathbb{V}_r[\boldsymbol{X}]}}, \quad \boldsymbol{X}_{\mathsf{Norm}} := \mathsf{Norm}(\boldsymbol{X}; \boldsymbol{\gamma}) = \boldsymbol{X}_{\mathrm{std}} \odot \boldsymbol{\gamma},$$

$$\mathsf{M} := \frac{\boldsymbol{X}_{\mathsf{Norm}} \boldsymbol{W}_Q \boldsymbol{W}_K^\top \boldsymbol{X}_{\mathsf{Norm}}^\top}{\sqrt{D}}, \quad \mathsf{A} := \mathrm{softmax}\left(\mathsf{M}\right), \quad \mathsf{S} := \mathsf{A}\boldsymbol{X}_{\mathsf{Norm}} \boldsymbol{W}_V \boldsymbol{W}_O,$$

$$\boldsymbol{Y} := \boldsymbol{X}^{(l-1/2)} = \boldsymbol{X} + \mathsf{S}.$$

We now compute $\frac{\partial \boldsymbol{Y}}{\partial \boldsymbol{W}_\bullet}$ and $\frac{\partial \boldsymbol{Y}}{\partial \boldsymbol{\gamma}}$:

$$\frac{\partial \boldsymbol{Y}}{\partial \boldsymbol{W}_Q} = \frac{\partial \mathsf{S}}{\partial \boldsymbol{W}_Q} = \frac{\partial \mathsf{S}}{\partial \mathsf{A}} \frac{\partial \mathsf{A}}{\partial \mathsf{M}} \frac{\partial \mathsf{M}}{\partial \boldsymbol{W}_Q} = \left( \boldsymbol{I}_n \otimes \boldsymbol{W}_O^\top \boldsymbol{W}_V^\top \boldsymbol{X}_{\mathsf{Norm}}^\top \right) \frac{\partial \mathsf{A}}{\partial \mathsf{M}} \left( \frac{\boldsymbol{X}_{\mathsf{Norm}} \otimes \boldsymbol{X}_{\mathsf{Norm}} \boldsymbol{W}_K}{\sqrt{D}} \right);$$

$$\frac{\partial \boldsymbol{Y}}{\partial \boldsymbol{W}_K} = \frac{\partial \mathsf{S}}{\partial \boldsymbol{W}_K} = \frac{\partial \mathsf{S}}{\partial \mathsf{A}} \frac{\partial \mathsf{A}}{\partial \mathsf{M}} \frac{\partial \mathsf{M}}{\partial \boldsymbol{W}_K} = \left( \boldsymbol{I}_n \otimes \boldsymbol{W}_O^\top \boldsymbol{W}_V^\top \boldsymbol{X}_{\mathsf{Norm}}^\top \right) \frac{\partial \mathsf{A}}{\partial \mathsf{M}} \left( \frac{\boldsymbol{X}_{\mathsf{Norm}} \otimes \boldsymbol{X}_{\mathsf{Norm}} \boldsymbol{W}_Q}{\sqrt{D}} \right);$$

$$\frac{\partial \boldsymbol{Y}}{\partial \boldsymbol{W}_V} = \frac{\partial \mathsf{S}}{\partial \boldsymbol{W}_V} = \mathsf{A} \boldsymbol{X}_{\mathsf{Norm}} \otimes \boldsymbol{W}_O^\top;$$

$$\frac{\partial \boldsymbol{Y}}{\partial \boldsymbol{W}_O} = \frac{\partial \mathsf{S}}{\partial \boldsymbol{W}_O} = \mathsf{A} \boldsymbol{X}_{\mathsf{Norm}} \boldsymbol{W}_V \otimes \boldsymbol{I}_D.$$

Moreover,

$$\frac{\partial \boldsymbol{Y}}{\partial \boldsymbol{\gamma}} = \frac{\partial \boldsymbol{Y}}{\partial \boldsymbol{X}_{\mathsf{Norm}}} \frac{\partial \boldsymbol{X}_{\mathsf{Norm}}}{\partial \boldsymbol{\gamma}} = \frac{\partial \mathsf{S}}{\partial \boldsymbol{X}_{\mathsf{Norm}}} \frac{\partial \boldsymbol{X}_{\mathsf{Norm}}}{\partial \boldsymbol{\gamma}}$$

$$= \left( \frac{1}{\sqrt{D}} \left( \boldsymbol{I}_n \otimes \boldsymbol{W}_O^\top \boldsymbol{W}_V^\top \boldsymbol{X}_{\mathsf{Norm}}^\top \right) \frac{\partial \mathsf{A}}{\partial \mathsf{M}} \left( \left( \boldsymbol{I}_n \otimes \boldsymbol{X}_{\mathsf{Norm}} \boldsymbol{W}_K \boldsymbol{W}_Q^\top \right) + \boldsymbol{K}_{n,n} \left( \boldsymbol{I}_n \otimes \boldsymbol{X}_{\mathsf{Norm}} \boldsymbol{W}_Q \boldsymbol{W}_K^\top \right) \right) \right.$$

$$\left. + \mathsf{A} \otimes \boldsymbol{W}_O^\top \boldsymbol{W}_V^\top \right) \left( \mathrm{diag} \big( \mathrm{vec}(\boldsymbol{X}_{\mathrm{std}}) \big) \big( \boldsymbol{1}_{n \times 1} \otimes \boldsymbol{I}_d \big) \right),$$

where $\boldsymbol{K}_{n,n}$ is the commutation matrix[5].

Recalling the proof in Appendix B.1, we have:

$$\| \boldsymbol{X}_{\mathrm{std}} \|_{\mathrm{F}} = \sqrt{nD}, \quad \| \boldsymbol{X}_{\mathsf{Norm}} \|_{\mathrm{F}} \le n \sqrt{D} \, \| \boldsymbol{\gamma} \|_{\mathrm{F}}.$$

Then, similar to the proof in Appendix B.1, we have the following upper bounds:

$$\left\| \frac{\partial \mathcal{Q}}{\partial \boldsymbol{W}_Q} \right\|_{\mathrm{F}} \le \frac{1}{\sqrt{D}} \left\| \frac{\partial \mathcal{Q}}{\partial \boldsymbol{Y}} \right\|_{\mathrm{F}} \left\| \frac{\partial \mathsf{A}}{\partial \mathsf{M}} \right\|_{\mathrm{F}} \| \boldsymbol{W}_K \|_{\mathrm{F}} \| \boldsymbol{W}_V \|_{\mathrm{F}} \| \boldsymbol{W}_O \|_{\mathrm{F}} \| \boldsymbol{X}_{\mathsf{Norm}} \|_{\mathrm{F}}^3$$

$$\le \frac{(n \sqrt{D})^3}{\sqrt{D}} \left\| \frac{\partial \mathcal{Q}}{\partial \boldsymbol{Y}} \right\|_{\mathrm{F}} \left\| \frac{\partial \mathsf{A}}{\partial \mathsf{M}} \right\|_{\mathrm{F}} \| \boldsymbol{W}_K \|_{\mathrm{F}} \| \boldsymbol{W}_V \|_{\mathrm{F}} \| \boldsymbol{W}_O \|_{\mathrm{F}} \| \boldsymbol{\gamma} \|_{\mathrm{F}}^3;$$

$$\left\| \frac{\partial \mathcal{Q}}{\partial \boldsymbol{W}_K} \right\|_{\mathrm{F}} \le \frac{1}{\sqrt{D}} \left\| \frac{\partial \mathcal{Q}}{\partial \boldsymbol{Y}} \right\|_{\mathrm{F}} \left\| \frac{\partial \mathsf{A}}{\partial \mathsf{M}} \right\|_{\mathrm{F}} \| \boldsymbol{W}_Q \|_{\mathrm{F}} \| \boldsymbol{W}_V \|_{\mathrm{F}} \| \boldsymbol{W}_O \|_{\mathrm{F}} \| \boldsymbol{X}_{\mathsf{Norm}} \|_{\mathrm{F}}^3$$

$$\le \frac{(n \sqrt{D})^3}{\sqrt{D}} \left\| \frac{\partial \mathcal{Q}}{\partial \boldsymbol{Y}} \right\|_{\mathrm{F}} \left\| \frac{\partial \mathsf{A}}{\partial \mathsf{M}} \right\|_{\mathrm{F}} \| \boldsymbol{W}_Q \|_{\mathrm{F}} \| \boldsymbol{W}_V \|_{\mathrm{F}} \| \boldsymbol{W}_O \|_{\mathrm{F}} \| \boldsymbol{\gamma} \|_{\mathrm{F}}^3;$$

$$\left\| \frac{\partial \mathcal{Q}}{\partial \boldsymbol{W}_V} \right\|_{\mathrm{F}} \le \left\| \frac{\partial \mathcal{Q}}{\partial \boldsymbol{Y}} \right\|_{\mathrm{F}} \| \mathsf{A} \|_{\mathrm{F}} \| \boldsymbol{W}_O \|_{\mathrm{F}} \| \boldsymbol{X}_{\mathsf{Norm}} \|_{\mathrm{F}} \le n \sqrt{D} \left\| \frac{\partial \mathcal{Q}}{\partial \boldsymbol{Y}} \right\|_{\mathrm{F}} \| \mathsf{A} \|_{\mathrm{F}} \| \boldsymbol{W}_O \|_{\mathrm{F}} \| \boldsymbol{\gamma} \|_{\mathrm{F}};$$

$$\left\| \frac{\partial \mathcal{Q}}{\partial \boldsymbol{W}_O} \right\|_{\mathrm{F}} \le \left\| \frac{\partial \mathcal{Q}}{\partial \boldsymbol{Y}} \right\|_{\mathrm{F}} \| \mathsf{A} \|_{\mathrm{F}} \| \boldsymbol{W}_V \|_{\mathrm{F}} \| \boldsymbol{X}_{\mathsf{Norm}} \|_{\mathrm{F}} \le n \sqrt{D} \left\| \frac{\partial \mathcal{Q}}{\partial \boldsymbol{Y}} \right\|_{\mathrm{F}} \| \mathsf{A} \|_{\mathrm{F}} \| \boldsymbol{W}_V \|_{\mathrm{F}} \| \boldsymbol{\gamma} \|_{\mathrm{F}};$$

$$\left\| \frac{\partial \mathcal{Q}}{\partial \boldsymbol{\gamma}} \right\|_{\mathrm{F}} \le \left\| \frac{\partial \mathcal{Q}}{\partial \boldsymbol{Y}} \right\|_{\mathrm{F}} \sqrt{n} \, \| \boldsymbol{X}_{\mathrm{std}} \|_{\mathrm{F}} \left( \frac{2}{\sqrt{D}} \left\| \left( \boldsymbol{I}_n \otimes \boldsymbol{W}_O^\top \boldsymbol{W}_V^\top \boldsymbol{X}_{\mathsf{Norm}}^\top \right) \frac{\partial \mathsf{A}}{\partial \mathsf{M}} \left( \boldsymbol{I}_n \otimes \boldsymbol{X}_{\mathsf{Norm}} \boldsymbol{W}_K \boldsymbol{W}_Q^\top \right) \right\|_{\mathrm{F}} + \left\| \mathsf{A} \otimes \boldsymbol{W}_O^\top \boldsymbol{W}_V^\top \right\|_{\mathrm{F}} \right)$$

$$\le n \sqrt{D} \left\| \frac{\partial \mathcal{Q}}{\partial \boldsymbol{Y}} \right\|_{\mathrm{F}} \left( \frac{2}{\sqrt{D}} \left\| \frac{\partial \mathsf{A}}{\partial \mathsf{M}} \right\|_{\mathrm{F}} \| \boldsymbol{W}_K \|_{\mathrm{F}} \| \boldsymbol{W}_Q \|_{\mathrm{F}} \| \boldsymbol{W}_V \|_{\mathrm{F}} \| \boldsymbol{W}_O \|_{\mathrm{F}} \| \boldsymbol{X}_{\mathsf{Norm}} \|_{\mathrm{F}}^2 + \| \mathsf{A} \|_{\mathrm{F}} \| \boldsymbol{W}_V \|_{\mathrm{F}} \| \boldsymbol{W}_O \|_{\mathrm{F}} \right)$$

$$\le \left\| \frac{\partial \mathcal{Q}}{\partial \boldsymbol{Y}} \right\|_{\mathrm{F}} \left( \frac{2(n \sqrt{D})^3}{\sqrt{D}} \left\| \frac{\partial \mathsf{A}}{\partial \mathsf{M}} \right\|_{\mathrm{F}} \| \boldsymbol{W}_K \|_{\mathrm{F}} \| \boldsymbol{W}_Q \|_{\mathrm{F}} \| \boldsymbol{W}_V \|_{\mathrm{F}} \| \boldsymbol{W}_O \|_{\mathrm{F}} \| \boldsymbol{\gamma} \|_{\mathrm{F}}^2 + n \sqrt{D} \| \mathsf{A} \|_{\mathrm{F}} \| \boldsymbol{W}_V \|_{\mathrm{F}} \| \boldsymbol{W}_O \|_{\mathrm{F}} \right).$$

---

[5]The commutation matrix $\boldsymbol{K}_{m,n}$ transforms column-wise vectorization into row-wise vectorization.

Therefore, if we define:

$$\Phi := \frac{(n\sqrt{D})^3}{\sqrt{D}} \left\|\frac{\partial \mathcal{Q}}{\partial \boldsymbol{Y}}\right\|_{\mathrm{F}} \left\|\frac{\partial \mathsf{A}}{\partial \mathsf{M}}\right\|_{\mathrm{F}} \|\boldsymbol{W}_K\|_{\mathrm{F}} \|\boldsymbol{W}_Q\|_{\mathrm{F}} \|\boldsymbol{W}_V\|_{\mathrm{F}} \|\boldsymbol{W}_O\|_{\mathrm{F}} \|\boldsymbol{\gamma}\|_{\mathrm{F}}^3,$$

$$\Psi := n\sqrt{D} \left\|\frac{\partial \mathcal{Q}}{\partial \boldsymbol{Y}}\right\|_{\mathrm{F}} \|\mathsf{A}\|_{\mathrm{F}} \|\boldsymbol{W}_V\|_{\mathrm{F}} \|\boldsymbol{W}_O\|_{\mathrm{F}} \|\boldsymbol{\gamma}\|_{\mathrm{F}},$$

then it holds that:

$$\left\|\frac{\partial \mathcal{Q}}{\partial \boldsymbol{W}_K}\right\|_{\mathrm{F}} \leq \frac{\Phi}{\|\boldsymbol{W}_K\|_{\mathrm{F}}}; \qquad \left\|\frac{\partial \mathcal{Q}}{\partial \boldsymbol{W}_Q}\right\|_{\mathrm{F}} \leq \frac{\Phi}{\|\boldsymbol{W}_Q\|_{\mathrm{F}}};$$

$$\left\|\frac{\partial \mathcal{Q}}{\partial \boldsymbol{W}_V}\right\|_{\mathrm{F}} \leq \frac{\Psi}{\|\boldsymbol{W}_V\|_{\mathrm{F}}}; \qquad \left\|\frac{\partial \mathcal{Q}}{\partial \boldsymbol{W}_O}\right\|_{\mathrm{F}} \leq \frac{\Psi}{\|\boldsymbol{W}_O\|_{\mathrm{F}}};$$

$$\left\|\frac{\partial \mathcal{Q}}{\partial \boldsymbol{\gamma}}\right\|_{\mathrm{F}} \leq \frac{2\Phi + \Psi}{\|\boldsymbol{\gamma}\|_{\mathrm{F}}}.$$

Therefore,

$$\mathcal{S}(\boldsymbol{W}_\bullet) = \frac{1}{\#(\boldsymbol{W}_\bullet)} \left\|\frac{\partial \mathcal{Q}}{\partial \boldsymbol{W}_\bullet}\right\|_{\mathrm{F}}^2 = \mathcal{O}\left(\frac{\Phi^2}{D^2 \|\boldsymbol{W}_\bullet\|_{\mathrm{F}}^2}\right), \quad \bullet \in \{K, Q\};$$

$$\mathcal{S}(\boldsymbol{W}_\bullet) = \frac{1}{\#(\boldsymbol{W}_\bullet)} \left\|\frac{\partial \mathcal{Q}}{\partial \boldsymbol{W}_\bullet}\right\|_{\mathrm{F}}^2 = \mathcal{O}\left(\frac{\Psi^2}{D^2 \|\boldsymbol{W}_\bullet\|_{\mathrm{F}}^2}\right), \quad \bullet \in \{V, O\};$$

$$\mathcal{S}(\boldsymbol{\gamma}) = \frac{1}{\#(\boldsymbol{\gamma})} \left\|\frac{\partial \mathcal{Q}}{\partial \boldsymbol{\gamma}}\right\|_{\mathrm{F}}^2 = \mathcal{O}\left(\frac{\Phi^2 + \Psi^2}{D \|\boldsymbol{\gamma}\|_{\mathrm{F}}^2}\right).$$

### B.3. Proof of Theorem 4.3

We focus on the transformation from $\boldsymbol{X}$ to $\boldsymbol{Y} := \mathsf{Norm}(\boldsymbol{X}\boldsymbol{W}_E; \boldsymbol{\gamma}^{(1/2)})$. For simplicity, we define:

$$\boldsymbol{Z} := \boldsymbol{X}\boldsymbol{W}_E, \quad \boldsymbol{Z}_{\mathrm{std}} := \frac{\boldsymbol{Z} - \mathbb{E}_r[\boldsymbol{Z}]}{\sqrt{\mathbb{Z}_r[\boldsymbol{Z}]}}, \quad \boldsymbol{Y} = \mathsf{Norm}(\boldsymbol{Z}; \boldsymbol{\gamma}) = \boldsymbol{Z}_{\mathrm{std}} \odot (\mathbf{1}_{n \times 1} \otimes \boldsymbol{\gamma}).$$

It is straightforward that:

$$\frac{\partial \boldsymbol{Y}}{\partial \boldsymbol{\gamma}} = \mathrm{diag}\big(\mathrm{vec}(\boldsymbol{Z}_{\mathrm{std}})\big)\big(\mathbf{1}_{n \times 1} \otimes \boldsymbol{I}_D\big).$$

Recalling the proof in Appendix B.1, we have:

$$\left\|\frac{\partial \boldsymbol{Y}}{\partial \boldsymbol{\gamma}}\right\|_{\mathrm{F}} \leq n\sqrt{D}.$$

Then we calculate $\frac{\partial \boldsymbol{Y}}{\partial \boldsymbol{W}_E}$. For simplicity, we denote

$$\tilde{\boldsymbol{Z}} := \boldsymbol{Z} - \mathbb{E}_r[\boldsymbol{Z}], \quad \boldsymbol{Z} = \begin{pmatrix} \tilde{\boldsymbol{z}}_1 \\ \dots \\ \tilde{\boldsymbol{z}}_d \end{pmatrix} \in \mathbb{R}^{d \times D},$$

$$\tilde{\boldsymbol{W}}_E := \boldsymbol{W}_E - \mathbb{E}_r[\boldsymbol{W}_E], \quad \boldsymbol{W}_E = \begin{pmatrix} \boldsymbol{w}_{E_1} \\ \dots \\ \boldsymbol{w}_{E_d} \end{pmatrix} \in \mathbb{R}^{d \times D}, \quad \tilde{\boldsymbol{W}}_E = \begin{pmatrix} \tilde{\boldsymbol{w}}_{E_1} \\ \dots \\ \tilde{\boldsymbol{w}}_{E_d} \end{pmatrix} \in \mathbb{R}^{d \times D}$$

By the proof in (Xiong et al., 2020), for a vector $\boldsymbol{x} \in \mathbb{R}^{1 \times D}$, denoted by $\tilde{\boldsymbol{x}} := \boldsymbol{x} - \mathbb{E}[\boldsymbol{x}]$, then $\frac{\partial \boldsymbol{x}_{\mathrm{std}}}{\partial \boldsymbol{x}} = \frac{\sqrt{D}}{\|\tilde{\boldsymbol{x}}\|_2} \left( \boldsymbol{I} - \frac{\tilde{\boldsymbol{x}}^\top \tilde{\boldsymbol{x}}}{\|\tilde{\boldsymbol{x}}\|_2^2} \right) \left( \boldsymbol{I} - \frac{1}{d} \mathbf{1}_{1 \times D}^\top \mathbf{1}_{1 \times D} \right)$. Thus, we have:

$$\frac{\partial \boldsymbol{Y}}{\partial \boldsymbol{W}_E} = \frac{\partial \boldsymbol{Y}}{\partial \boldsymbol{Z}_{\mathrm{std}}} \frac{\partial \boldsymbol{Z}_{\mathrm{std}}}{\partial \boldsymbol{Z}} \frac{\partial \boldsymbol{Z}}{\partial \boldsymbol{W}_E}$$

$$= (\boldsymbol{I}_n \otimes \operatorname{diag}(\operatorname{vec}(\boldsymbol{\gamma}))) \operatorname{diag}\left( \left\{ \frac{\sqrt{D}}{\|\tilde{\boldsymbol{z}}_i\|_2} \left( \boldsymbol{I} - \frac{\tilde{\boldsymbol{z}}_i^\top \tilde{\boldsymbol{z}}_i}{\|\tilde{\boldsymbol{z}}_i\|_2^2} \right) \left( \boldsymbol{I} - \frac{1}{D} \mathbf{1}_{1 \times D}^\top \mathbf{1}_{1 \times D} \right) \right\}_{i \in [n]} \right) (\boldsymbol{X} \otimes \boldsymbol{I}_D).$$

Recalling the relationship $z_{i,j} = \sum_{k=1}^d x_{i,k} w_{k,j}$, we have $\mathbb{E}[\boldsymbol{z}_i] = \sum_{k=1}^d x_{i,k} \mathbb{E}[\boldsymbol{w}_k]$, which implies

$$\tilde{\boldsymbol{z}}_i = \sum_{k=1}^d x_{i,k} \tilde{\boldsymbol{w}}_k.$$

Combining this property with the that are one-hot fact of the inputs $\boldsymbol{X}$, we have:

$$\min_{i \in [n]} \|\tilde{\boldsymbol{z}}_i\|_2 \geq \min_{k \in [d]} \|\tilde{\boldsymbol{w}}_k\|_2.$$

Additionally, the one-hot encoding ensures:

$$\|\boldsymbol{X}\|_{\mathrm{F}} = \left( \sum_{i=1}^n x_{i,j}^2 \right)^{1/2} = \sqrt{n}.$$

Now we have the following bound:

$$\left\| \frac{\partial \boldsymbol{Y}}{\partial \boldsymbol{W}_E} \right\|_{\mathrm{F}}$$

$$\leq \|\boldsymbol{I}_n \otimes \operatorname{diag}(\operatorname{vec}(\boldsymbol{\gamma}))\|_{\mathrm{F}} \left\| \operatorname{diag}\left( \left\{ \frac{\sqrt{D}}{\|\tilde{\boldsymbol{z}}_i\|_2} \left( \boldsymbol{I} - \frac{\tilde{\boldsymbol{z}}_i^\top \tilde{\boldsymbol{z}}_i}{\|\tilde{\boldsymbol{z}}_i\|_2^2} \right) \left( \boldsymbol{I} - \frac{1}{D} \mathbf{1}_{1 \times D}^\top \mathbf{1}_{1 \times D} \right) \right\}_{i \in [n]} \right) \right\|_2 \|\boldsymbol{X} \otimes \boldsymbol{I}_D\|_2$$

$$\leq \sqrt{n} \|\boldsymbol{\gamma}\|_{\mathrm{F}} \frac{\sqrt{D}}{\min_{i \in [n]} \|\tilde{\boldsymbol{z}}_i\|_2} \|\boldsymbol{X}\|_2 \leq n\sqrt{D} \frac{\|\boldsymbol{\gamma}\|_{\mathrm{F}}}{\min_{i \in [n]} \|\tilde{\boldsymbol{z}}_i\|_2} \leq n\sqrt{D} \frac{\|\boldsymbol{\gamma}\|_{\mathrm{F}}}{\min_{i \in [d]} \|\tilde{\boldsymbol{w}}_i\|_2}.$$

If we choose $\Psi := n\sqrt{D} \|\boldsymbol{\gamma}\|_{\mathrm{F}}$, then we have:

$$\left\| \frac{\partial \boldsymbol{Y}}{\partial \boldsymbol{\gamma}} \right\|_{\mathrm{F}} \leq \frac{\Psi}{\|\boldsymbol{\gamma}\|_{\mathrm{F}}}, \quad \left\| \frac{\partial \boldsymbol{Y}}{\partial \boldsymbol{W}_E} \right\|_{\mathrm{F}} \leq \frac{\Psi}{\min_{i \in [d]} \|\tilde{\boldsymbol{w}}_i\|_2}.$$

Therefore,

$$\mathcal{S}(\boldsymbol{W}_E) = \frac{1}{\#(\boldsymbol{W}_E)} \left\| \frac{\partial \mathcal{Q}}{\partial \boldsymbol{W}_E} \right\|_{\mathrm{F}}^2 = \mathcal{O}\left( \frac{\Psi^2}{Dd \min_{i \in [d]} \|\tilde{\boldsymbol{w}}_i\|_2^2} \right);$$

$$\mathcal{S}(\boldsymbol{\gamma}) = \frac{1}{\#(\boldsymbol{\gamma})} \left\| \frac{\partial \mathcal{Q}}{\partial \boldsymbol{\gamma}} \right\|_{\mathrm{F}}^2 = \mathcal{O}\left( \frac{\Psi^2}{D \|\boldsymbol{\gamma}\|_{\mathrm{F}}^2} \right).$$

