# OpenReview forum: "The Sharpness Disparity Principle in Transformers for Accelerating Language Model Pre-Training"
_ICML.cc/2025/Conference — ICML 2025 poster_

### Official Review · Reviewer_w3eV · 2025-03-10

**Overall Recommendation:** 4

**Summary:**

This paper uncovers a sharpness disparity across different blocks in Transformers, which persists throughout the training process. The authors propose a novel Blockwise Learning Rate strategy to accelerate LLM (e.g., GPT and LLaMA) pre-training. Furthermore, the proposed method consistently achieves lower loss.

**Claims And Evidence:**

Yes.

**Essential References Not Discussed:**

NA.

**Experimental Designs Or Analyses:**

Yes, I think the experiments are solid enough to support the authors' claim.

**Methods And Evaluation Criteria:**

Yes.

**Other Comments Or Suggestions:**

NA.

**Other Strengths And Weaknesses:**

Strengths:
1. The paper is written clearly.
2. The paper designs a systematic study on the impact of sharpness in LLM pretraining, and the experiments support their claims well.
3. The experimental results are promising.

Weaknesses:
1. The approximation is only performed for the digonal Hessian matrix, and the gap between the estimated Hessian matrix and the true Hessian matrix is not been controlled.
2. For non-LLM tasks and other optimizers, the sharpness principle may not hold and need many computations to check.

**Questions For Authors:**

1. Why do you not decrease the LRs along high-sharpness directions?

**Relation To Broader Scientific Literature:**

NA.

**Theoretical Claims:**

I did not check the proof carefully, but the theoretical results seem faithful and supported by the empirical results.

---

> ### Author Rebuttal · Authors · 2025-04-01
>
> Thank you for your great efforts on the review of this paper and your appreciation. We will try our best to address your questions.
>
> **Q1: Concerns about the gap between estimated digonal Hessian and the true Hessian.** "The approximation is only performed for the diagonal Hessian matrix, and the gap between the estimated Hessian matrix and the true Hessian matrix is not been controlled."
>
> **A1**: Thanks for this question.
> - First, we’d like to clarify that our sharpness measure (Eq. (4)) is based on the trace of blockwise Hessians, for which *only the diagonal Hessians are needed*. This follows from the identity ${\rm Tr}(H)={\rm Tr}({\rm diag}(H))$, which we will clarify in the revised version.
> - To approximate the diagonal Hessian, we adopt the diagonal Fisher matrix. The Fisher is widely regarded as a reasonable approximation of the Hessian in optimization and deep learning [LeCun et al., 1998; Amari, 1998; Martens, 2014].
>
> **Q2: Suggestions for experiments on non-LLM tasks and other optimizers.** "For non-LLM tasks and other optimizers, the sharpness principle may not hold and need many computations to check."
>
>
> **A2**: Thanks for the constructive suggestions. To address them, we conducted two **new experiments**.
> - **Other optimizers.** We evaluated the sharpness disparity principle on LLaMA trained with Lion [Chen et al., 2023]. As shown in [Fig.R7](https://anonymous.4open.science/api/repo/ICML-2025-Sharpness-OPT-V2/file/R7_law_lion.pdf?v=c52ad373), it exhibits *almost the same* principle as with AdamW (Eq. (1) and Fig 3(b)). Then we integrated Blockwise LR into Lion. Remarkably, as shown in [Fig.R6](https://anonymous.4open.science/api/repo/ICML-2025-Sharpness-Optimization/file/R6_lion_web.pdf?v=6932b09d), Lion with Blockwise LR achieves lower terminal loss and 2x speedup over well-tuned Lion.
> - **Non-LLM tasks.** While our primary focus is on language models, as indicated by the title, we followed your suggestion and evaluated the sharpness principle on a ViT-B trained on ImageNet-1k. Surprisingly, as shown in [Fig. R8](https://anonymous.4open.science/api/repo/ICML-2025-Sharpness-OPT-V2/file/R8_ViT.pdf?v=e754a2fa), ViT exhibits a similar sharpness ordering as LLMs: S(QK)<S(FFN)<S(VO)<S(Norm). The only difference is that the embedding layer is no longer the flattest, likely due to structural differences between image and language inputs. These results suggest Blockwise LR can be extended to vision models using this revised principle.
>
>
> **Q3: Questions on the design of Blockwise LR.** "Why do you not decrease the LRs along high-sharpness directions?"
>
> **A3**: Thanks for this insightful question.
> - Our design follows the view in [Wen et al., 2024; Wang et al., 2024; Song et al., 2024]: low-sharpness directions primarily drive loss descent, while high-sharpness directions determine training stability. To maintain stability, we keep the learning rate in high-sharpness directions unchanged.
> - Reducing LR in high-sharpness directions may suppress oscillations in these directions but alters the stability condition, and its long-term impact remains unclear. [Wen et al., 2024] shows that using relatively large LR in high-sharpness ("hill") directions early in training can result in lower final loss.
>
> ### Reference
>
> Allen-Zhu et al. A Convergence Theory for Deep Learning via Over-Parameterization. 2018.
>
> Amari, S. Natural Gradient Works Efficiently in Learning. Neural Computation. 1998.
>
> Chen et al. Symbolic Discovery of Optimization Algorithms. 2023.
>
> D'Angelo et al. Why Do We Need Weight Decay in Modern Deep Learning? 2023.
>
> Du et al. Understanding Emergent Abilities of Language Models from the Loss Perspective. 2024.
>
> Du et al. Gradient Descent Finds Global Minima of Deep Neural Networks. 2018.
>
> Hoffmann et al. Training Compute-Optimal Large Language Models. 2022.
>
> Hu et al. MiniCPM: Unveiling the Potential of Small Language Models with Scalable Training Strategies. 2024.
>
> LeCun et al. Efficient Backprop.1998.
>
> Liu et al. Sophia: A scalable stochastic second-order optimizer for language model pre-training. 2024.
>
> Martens, J. New insights and perspectives on the natural gradient method. 2014.
>
> Song et al. Does SGD really happen in tiny subspaces? 2024.
>
> Wang et al. Improving Generalization and Convergence by Enhancing Implicit Regularization. 2024.
>
> Wen et al. Understanding Warmup-Stable-Decay Learning Rates: A River Valley Loss Landscape Perspective. 2024.

---

> > ### Comment · Reviewer_w3eV · 2025-04-05
> >
> > I have improved my score from 3 to 4.

---

> > > ### Author Response · Authors · 2025-04-08
> > >
> > > Thank you sincerely for taking your time to revisit your evaluation and improve your score.

---

### Official Review · Reviewer_jDEo · 2025-03-13

**Overall Recommendation:** 4

**Summary:**

The authors demonstrate that there is a sharpness disparity between the different transformer blocks, which appears early in training and persists throughout. Based on their observation, the authors introduce a novel approach called Blockwise Learning Rate, which adjusts the learning rate of each transformer block based on its sharpness. Blockwise Learning Rate with AdamW achieves lower loss with the same number of gradient steps, and reaches the same loss with nearly half the steps on various sizes of GPT-2 and LLaMA across two datasets. Additionally, the authors demonstrate that Blockwise Learning Rate is compatible with Adam-mini, a memory-efficient variant of Adam.

**Claims And Evidence:**

The two claims are (1) there is a sharpness disparity between the different transformer blocks, and (2) adjusting the learning independently for each block based on its sharpness results in faster training speed with respect to the number of gradient steps. The theoretical proofs and experimental evaluation support their claims.

**Essential References Not Discussed:**

I am not aware of missing relevant works.

**Experimental Designs Or Analyses:**

As mentioned above, the experimental design is sound. The authors consider two widely popular models, GPT-2 and LLAMA, with various sizes. The two datasets considered are well established too, although they are small in comparison to modern datasets such as RedPajama or RefineWeb. The training procedure is modern, relying on AdamW with tuned β=(0.9,0.95) and a warmup phase followed by a cosine decay. In the literature, it is more common for the final learning rate to be 10% of the peak learning rate instead of 5%, but I do not believe that this invalidates their observations. The number of gradient steps is relatively small (30K to 100K) compared to modern LLMs, but I believe it is just enough to draw conclusions. The performance is only evaluated in terms of loss, which may not necessarily translate to better performance on downstream tasks, so I would like to see some downstream evaluation to be added.

**Methods And Evaluation Criteria:**

The evaluation methodology is sound: the models, datasets, and training procedure are standard, and they consider various model sizes, optimizers, and hyperparameters. However, the authors only report the loss and do not evaluate the downstream performance, which would strengthen their evidence.

**Other Comments Or Suggestions:**

I suggest replacing "point-wise feed-forward network" with either "position-wise" or "token-wise" as they are more common.

**Other Strengths And Weaknesses:**

The figures are clear, especially how the sharpness of each block is depicted in Figure 3. I appreciate that the authors validated their proposed Blockwise Learning Rate across two models, two datasets, two optimizers, and various sizes. Additionally, I appreciate that they tuned the learning rate to ensure optimal performance for AdamW.

The main weaknesses are the lack of downstream evaluation, which may reveal a closer performance than the loss suggests, and the number of gradient steps, which may close the gap.

**Questions For Authors:**

- Can you extend the training of one of the small models, preferably Llama 0.25B?
- Can you evaluate a few models on downstream benchmarks?
- You have shown that the Blockwise Learning Rate works across optimizers. Do you know if it also works across learning rate schedulers? Can you apply your Blockwise Learning Rate with warm-stable-decay (WSD) instead of cosine decay?

**Relation To Broader Scientific Literature:**

This paper contributes to the field of sharpness analysis and efficient optimizers for transformers. As far as I am aware, this is the firs study to consider the sharpness of blocks rather than layers and to suggest modifying the learning rate per-block across the entire model.

**Theoretical Claims:**

I have not reviewed the proofs in detail due to a lack of time. However, the demonstrations in the main paper *appear* to be sound and supported by empirical evidence.

---

> ### Author Rebuttal · Authors · 2025-04-01
>
> Thank you for your great efforts on the review of this paper and your appreciation. We will try our best to address your questions.
>
> **Q1: Concerns about the downstream performance.** "However, the authors only report the loss and do not evaluate the downstream performance, which would strengthen their evidence." "The performance is only evaluated in terms of loss, which may not necessarily translate to better performance on downstream tasks, so I would like to see some downstream evaluation to be added."
>
> **A1**: Thanks for raising this concern.
> - We clarify that In LLM pretraining, downstream performance is widely known to correlate strongly with final pretraining loss and less so with factors like architecture or optimizer choice. See [Du et al., 2024] for a detailed analysis. Thus, the primary focus in LLM pre-training is to reduce the final loss as much as possible, under specific compute and data budgets.
> - In response to your suggestion, we **evaluated downstream performance**. As shown in [Tab.R4](https://anonymous.4open.science/api/repo/SharpnessTable-Downstream/file/R4_evaluate.pdf?v=8dadc1ea), LLaMA trained with our algorithm outperforms the one trained with AdamW across all evaluated tasks.
>
> **Q2: Concerns about the final learning rate.** "In the literature, it is more common for the final learning rate to be 10% of the peak learning rate instead of 5%, but I do not believe that this invalidates their observations."
>
> **A2**: Thank you for the helpful comment.
> - In our experiments, we followed the setup in Sophia  (Sec. 3.1 in [Liu et al., 2023]), where the final LR is set to 5% of the peak LR.
> - As noted by the reviewer, it is more common in the literature for the final LR to be 10% of the peak LR. In our **new experiments** conducted on C4 dataset, we adopt this setting, and the corresponding results are shown in [Fig.R2](https://anonymous.4open.science/api/repo/ICML-2025-Sharpness-Optimization/file/R2_scaling_c4.pdf?v=f2e12859).
>
> **Q3: Concerns about the number of training steps.** "The number of gradient steps is relatively small (30K to 100K) compared to modern LLMs, but I believe it is just enough to draw conclusions." "Can you extend the training of one of the small models, preferably Llama 0.25B?"
>
> **A3**: Thanks for this question.
> - We clarify that our training steps are sufficiently large given current model and dataset sizes. For example, 100k steps on OpenWebText yields 480x1024x100k~**50 billion tokens**. According to Tab. 3 in [Hoffmann et al., 2022], most of our experiments *exceed the recommended token budget*.
> - Moreover, this training setup aligns with standard practice in the community, e.g., [Liu et al., 2023; D'Angelo et al., 2023].
> - Following your suggestion, we conducted a **new experiment** by extending the training of 0.25B LLaMA from 50k/100k to 150k/300k steps. As shown in [Fig.R3](https://anonymous.4open.science/api/repo/ICML-2025-Sharpness-Optimization/file/R3_025B_long.pdf?v=16de8e07), Blockwise LR still achieves lower terminal loss and is 2x faster than AdamW even at longer durations.
>
>
> **Q4: Suggestions for "point-wise feed-forward network".** "I suggest replacing "point-wise feed-forward network" with either "position-wise" or "token-wise" as they are more common."
>
> **A4**: Thank you for your suggestion. We will revise the terminology in the revised version.
>
>
> **Q5: Suggestions for WSD experiments.** "Do you know if it also works across learning rate schedulers? Can you apply your Blockwise Learning Rate with warm-stable-decay (WSD) instead of cosine decay?"
>
> **A5**: Thanks for the constructive suggestion. Following your recommendation, we conducted a **new experiment** using the WSD scheduler. As shown in [Fig.R5](https://anonymous.4open.science/api/repo/ICML-2025-Sharpness-Optimization/file/R5_wsd_web.pdf?v=f591fc57), Blockwise LR still achieves a 2x speedup over AdamW under this setting.
>
>
> ### Reference
> Due to space limit, see Reference in *Rebuttal to Reviewer w3eV*.

---

> > ### Comment · Reviewer_jDEo · 2025-04-03
> >
> > I appreciate the reviewers' clarifications and am pleased with the additional experiments. I have updated my score accordingly.

---

> > > ### Author Response · Authors · 2025-04-08
> > >
> > > We're glad that our responses addressed your concerns and appreciate your willingness to revise your score accordingly.

---

### Official Review · Reviewer_qPBA · 2025-03-16

**Overall Recommendation:** 3

**Summary:**

The paper proposed a blockwise learning rate method to accelerate training. The blockwise learning rate is designed based on blockwise sharpness estimation.  The writing is clear. The principle is reasonable and makes sense to me. The experiments are mostly convincing. Overally speaking, this is a good paper and might motivate more future algorithm designs.  The paper is worth sharing with community.

**Claims And Evidence:**

yes

**Essential References Not Discussed:**

no

**Experimental Designs Or Analyses:**

yes

**Methods And Evaluation Criteria:**

yes

**Other Comments Or Suggestions:**

see below

**Other Strengths And Weaknesses:**

see below

**Questions For Authors:**

Q1: In Figure 4 and 5, what do "(50k)" and "(100k)" mean? I cannot find the description anywhere around the figure or in Section 6.

Q2: According to Figure 4, blockwise lr (50k) converges slower than AdamW (50k) in the early stage, and then  blockwise lr (50k) converges faster in the final steps. Why would this happen? Any explanation or intuition?

Q3: In Figure 4, the training seems far from convergence. What is the total number of tokens T?   Does the advantage of blockwise lr maintains if we train more tokens?

Q4: The current blockwise lr is tested on the top of cosine schedule. How does the proposed blockwise lr reconcile with other lr schedules like WSD? Does the acceleration maintains?

**Relation To Broader Scientific Literature:**

no

**Theoretical Claims:**

yes

---

> ### Author Rebuttal · Authors · 2025-04-01
>
> Thank you for your great efforts on the review of this paper and your appreciation. We will try our best to address your questions.
>
> **Q1: Questions of the meaning of "(50k)" and "(100k)".** "In Figure 4 and 5, what do "(50k)" and "(100k)" mean? I cannot find the description anywhere around the figure or in Section 6."
>
> **A1**: Thank you for pointing this out. 50k and 100k refers to the total training steps. We will provide more explanations in future revision.
>
> **Q2: Questions about the faster convergence of Blockwise LR in the final steps.** "According to Figure 4, blockwise lr (50k) converges slower than AdamW (50k) in the early stage, and then blockwise lr (50k) converges faster in the final steps. Why would this happen? Any explanation or intuition?"
>
> **A2**: Very interesting question. This behavior resembles that of WSD schedulers, which often outperform cosine decay in later stages. A preliminary explanation draws on the **river-valley** loss landscape [Wen et al., 2024], which splits the loss into two components: a *river component* (the primary loss along the river at the bottom of the hills) and a *hill component* (additional loss from deviations in height from the river’s course).
> - *Early training*: Blockwise LR boosts LR in river (low-sharpness) directions, enabling faster core progress minimizing the river component,  but also inevitably in some hill (high-sharpness) directions due to noise in data or Hessian estimates—causing larger oscillations and higher loss.
> - *Late training*: As LR decays, the oscillations along the hill component diminish and iterates settle close to the river path [Wen et al., 2024]. Since Blockwise LR made more progress along the river early on, it achieves a lower terminal loss.
>
> We will add this discussion in the revision.
>
> **Q3: Questions about the total number of training tokens.** "In Figure 4, the training seems far from convergence. What is the total number of tokens T? Does the advantage of blockwise lr maintains if we train more tokens?"
>
> **A3**: Thanks for this question.
> - We clarify that our training steps are sufficiently large given current model and dataset sizes. For example, 100k steps on OpenWebText yields 480x1024x100k~*50 billion tokens*. According to Tab. 3 in [Hoffmann et al., 2022], most of our experiments *exceed the recommended token budget*.
> - Moreover, this training setup aligns with standard practice in the community, e.g., [Liu et al., 2023; D'Angelo et al., 2023].
> - Following your suggestion, we conducted a **new experiment** by extending the training of 0.25B LLaMA from 50k/100k to 150k/300k steps. As shown in [Fig.R3](https://anonymous.4open.science/api/repo/ICML-2025-Sharpness-Optimization/file/R3_025B_long.pdf?v=16de8e07), Blockwise LR still achieves lower terminal loss and is 2x faster than AdamW even at longer durations.
>
>
> **Q4: Suggestions for WSD experiments.** "The current blockwise lr is tested on the top of cosine schedule. How does the proposed blockwise lr reconcile with other lr schedules like WSD? Does the acceleration maintains?"
>
> **A4**: Thanks for the constructive suggestion. Following your recommendation, we conducted a **new experiment** using the WSD scheduler. As shown in [Fig.R5](https://anonymous.4open.science/api/repo/ICML-2025-Sharpness-Optimization/file/R5_wsd_web.pdf?v=f591fc57), Blockwise LR still achieves a 2x speedup over AdamW under this setting.
>
>
> ### Reference
> Due to space limit, see Reference in *Rebuttal to Reviewer w3eV*.

---

> > ### Comment · Reviewer_qPBA · 2025-04-03
> >
> > I would like to thank the authors for the rebuttal.  The new experiments in the rebuttal seem convincing.
> >
> > A kind suggestion for the authors to revise the paper: In the future version, I suggest the author align all the figures in the paper (which uses larger model size ~1B) with the standard of the new figures in the rebuttal (which uses small model size ~0.25B).  The figures in the rebuttal seem more professional and convincing, perhaps due to the longer training durations or due to certain smoothing. In comparison, the current figures in the script seem rather preliminary and noisy. It is not easy to make fair judgments based on the figures in the current script.

---

> > > ### Author Response · Authors · 2025-04-04
> > >
> > > Thank you for your kind feedback. The figures in the rebuttal are more smoothed due to larger training steps and multi-node evaluation. By employing multi-node evaluation, we enlarge the data available for each evaluation, resulting in smoother curves. However, for the figures in the main paper, we follow the experimental settings of nanoGPT (Karpathy, A., 2022) and Sophia (Liu et al., 2024) and use only single-node evaluation. **In our future revision, we will follow your suggestion to align the figures in the main paper with the standard of figures in the rebuttal**.
> > >
> > > We hope our response has adequately addressed your concerns, and we would greatly appreciate it if you could reconsider the final assessment in light of the clarifications and resolutions we have provided.
> > >
> > > ### Reference
> > > Karpathy, A. nanoGPT. https://github.com/karpathy/nanoGPT. GitHub repository, 2022.

---

### Official Review · Reviewer_umnV · 2025-03-17

**Overall Recommendation:** 3

**Summary:**

This paper presents the Sharpness Disparity Principle, which identifies a systematic difference in sharpness across different transformer components. Specifically, the authors find that normalization layers exhibit the highest sharpness, while embedding layers have the lowest, with other blocks lying in between. This pattern emerges early in training and persists throughout.

Leveraging this observation, they propose Blockwise LR, a novel learning rate scaling strategy that assigns different LRs to different block types based on their sharpness. By integrating this strategy into AdamW, they achieve nearly 2x training speed-ups while maintaining (or improving) final loss values across different models (from 0.12B to 1.1B).

**Claims And Evidence:**

Most calims are supported clearly. But I have some concerns:
 - The results focus on smaller-scale pretraining (1.1B max). It remains unclear whether this strategy generalizes to massive-scale models.
- The cause of this disparity is not deeply analyzed. While the paper suggests that parameter norms influence sharpness, a more in-depth theoretical analysis would be helpful.
- The number of training steps is not that large. It is better to observe from a larger time scale.

**Essential References Not Discussed:**

NA

**Experimental Designs Or Analyses:**

Pros:
- Results are consistent across different datasets and model sizes.
- Ablation studies show that modifying LR for Norm layers degrades performance.

Cons:
- Most experiments stop at 100K steps. Does the speedup persist in billion-step training?
- The study focuses on pretraining loss but does not analyze downstream performance on reasoning/QA tasks.

**Methods And Evaluation Criteria:**

Yes. Blockwise LR is well-motivated by empirical findings and aligns with existing work on learning rate adaptation. Ablation studies clearly show that increasing LR for norm layers is harmful, supporting the rationale behind the method.

Cons:
- Limited discussion on transferability to larger models. Will Blockwise LR still provide speedups at large scale?
- Lack of comparisons to alternative optimization strategies. It would be useful to compare against layerwise LR tuning methods.

**Other Comments Or Suggestions:**

NA

**Other Strengths And Weaknesses:**

Would benefit from larger-scale testing and comparisons to other adaptive optimizers.

**Questions For Authors:**

- What is the precision setting? Is it trained based on mix-precision?
- Can this be combined with second-order optimizers like Sophia or Shampoo?

**Relation To Broader Scientific Literature:**

The work is well-grounded.

**Theoretical Claims:**

Yes. The paper provides some theoretical justification by analyzing blockwise sharpness via the Fisher Information Matrix.

---

> ### Author Rebuttal · Authors · 2025-03-31
>
> Thank you for your great efforts on the review of this paper and your appreciation. We'll try our best to address your questions.
>
> **Q1: Suggestion for larger-scale experiments.**
>
> **A1**: Thanks for this question.
> - We clarify that our largest model (1.1B) is large enough for current dataset scale (OpenWebText and MiniPile), consistent with standard practices in the community (e.g., [Liu et al, 2023] trains models up to 0.77B on OpenWebText).
> - Our method scales well from 0.12B to 1.1B models, suggesting its potential scalability (see [Fig.R1](https://anonymous.4open.science/api/repo/ICML-2025-Sharpness-Optimization/file/R1_scaling_web.pdf?v=5c61cfe0). We are currently extending our experiments to **2B models**; the results are on the way.
> - To further support the effectiveness, we conducted **additional experiments** on a larger dataset, C4. In [Fig.R2](https://anonymous.4open.science/api/repo/ICML-2025-Sharpness-Optimization/file/R2_scaling_c4.pdf?v=f2e12859), our algorithm consistently achieves lower terminal loss than AdamW across model sizes from 0.2B to 1B, consistent with previous findings.
>
> **Q2: Concerns about the deep cause of the sharpness disparity principle.**
>
> **A2**: Thanks for this question.
> - We clarify that we have provided a first attempt to analyze this sharpness disparity principle from the parameter norms, supported by experiments and discussion following the theorems. Moreover, the discussion in Lines 298–306 offers an intuitive explanation from the multiplicative structure of Transformer blocks.
> - A more rigorous theoretical analysis is beyond this paper’s scope. Even for simpler two-layer neural networks, the analysis of the parameter norms remains challenging [Du et al, 2018; Allen-Zhu et al, 2018]. We leave this as important future work.
>
> **Q3: Suggestion for training steps.**
>
> **A3**: Thanks for this question.
> - We clarify that our training steps are sufficiently large given current model and dataset sizes. For example, 100k steps on OpenWebText yields 480x1024x100k~**50 billion tokens**. According to Tab. 3 in [Hoffmann et al, 2022], most of our experiments *exceed the recommended token budget*.
> - Moreover, this training setup aligns with standard practice in the community, e.g., [Liu et al, 2023; D’Angelo, 2023].
> - Following your suggestion, we conducted a **new experiment** by extending the training of 0.25B LLaMA from 50k/100k to 150k/300k steps. In [Fig.R3](https://anonymous.4open.science/api/repo/ICML-2025-Sharpness-Optimization/file/R3_025B_long.pdf?v=16de8e07), Blockwise LR still achieves lower terminal loss and is 2x faster than AdamW even at longer durations.
>
> **Q4: Concerns about downstream performance.**
>
> **A4**: Thanks for raising this concern.
> - We clarify that In LLM pretraining, downstream performance is widely known to correlate strongly with final pretraining loss and less so with factors like architecture or optimizer choice. See [Du et al, 2024] for a detailed analysis. Thus, the primary focus in LLM pre-training is to reduce the final loss as much as possible, under specific compute and data budgets.
> - In response to your suggestion, we **evaluated downstream performance**. In [Tab.R4](https://anonymous.4open.science/api/repo/SharpnessTable-Downstream/file/R4_evaluate.pdf?v=8dadc1ea), LLaMA trained with our algorithm outperforms the one trained with AdamW across all evaluated tasks.
>
> **Q5: Concerns about comparison with other optimization strategies or optimizers.**
>
> **A5**: Thanks for the constructive question.
> - We clarify that our Blockwise LR is the first successful blockwise LR method for Transformers. As noted in Remark 1.1, traditional layerwise LR methods--originally developed for MLPs and CNNs--have not translated successfully to Transformers.
> - Notably, Blockwise LR is compatible with various optimizers, e.g., AdamW and Adam-mini. Thus, instead of comparing with other optimizers, we follow your suggestions (in Q7) to evaluate Blockwise LR in combination with other optimizers (details in A7).
> - Following your suggestion, we also conducted a **new experiment** using another popular strategy, WSD scheduler [Hu et al, 2024]. In [Fig.R5](https://anonymous.4open.science/api/repo/ICML-2025-Sharpness-Optimization/file/R5_wsd_web.pdf?v=f591fc57), Blockwise LR still achieves a 2x speedup over AdamW in this setting.
>
> **Q6: Question about the precision settings.**
>
> **A6**: All the models are trained with BFloat16. We will clarify it in the revised version.
>
> **Q7: Question about the compatibility with other optimizers.**
>
> **A7**: Thanks for the interesting question. We followed your suggestion and combined Blockwise LR with another popular optimizer, Lion [Chen et al, 2023]. In [Fig.R6](https://anonymous.4open.science/api/repo/ICML-2025-Sharpness-Optimization/file/R6_lion_web.pdf?v=6932b09d), this combination achieves lower terminal loss and 2x speedup over well-tuned Lion.
>
> ### Reference
> Due to space limit, see Reference in *Rebuttal to Reviewer w3eV*.

---

### Decision · Program_Chairs · 2025-05-01

**Decision:**

Accept (poster)

**Comment:**

Reviewers agree that the idea on measure sharpness of different layers of the Transformer, and use it to change LR is interesting. The experiments are also quite thorough. Moreover, the training dynamics (slower than Adam first and faster later) also triggers discussions. Overall the work is worth publishing and may draw interest in the community.